# FLOORPLANQA: A BENCHMARK FOR SPATIAL REASONING IN LLMS USING STRUCTURED REPRESENTATIONS

## ABSTRACT

We introduce FloorplanQA, a diagnostic benchmark for evaluating spatial reasoning in large-language models (LLMs). FloorplanQA is grounded in structured representations of indoor scenes, such as (e.g., kitchens, living rooms, bedrooms, bathrooms, and others), encoded symbolically in JSON or XML layouts. The benchmark covers core spatial tasks, including distance measurement, visibility, path finding, and object placement within constrained spaces. Our results across a variety of frontier open-source and commercial LLMs reveal that while models may succeed in shallow queries, they often fail to respect physical constraints, preserve spatial coherence, though they remain mostly robust to small spatial perturbations. FloorplanQA uncovers a blind spot in today's LLMs: inconsistent reasoning about indoor layouts. We hope this benchmark inspires new work on language models that can accurately infer and manipulate spatial and geometric properties in practical settings.

## 1 INTRODUCTION

Recent progress in large language models (LLMs) has revealed strong capabilities in structured reasoning, yet spatial inference over plausible, physically feasible environments such as indoor layouts remains poorly understood. In numerous practical applications, including architectural design, assistive planning, and embodied interaction, spatial understanding is handled through structured formats such as JSON, in which objects are specified by position, size, and orientation, rather than through images or natural language. Reasoning in these contexts requires geometric inference over symbolic layouts, not pixel-level perception.

We introduce **FloorplanQA**, a benchmark to evaluate spatial reasoning in LLMs using 2D floorplans represented in structured text-based formats. Each instance consists of a JSON-encoded layout paired with natural language questions that require the model to compute distances, evaluate placement feasibility, assess visibility, and reason about spatial constraints. FloorplanQA isolates symbolic spatial reasoning over inputs that mirror the abstractions used by designers, architects, and agents operating in structured environments.

Although LLMs can increasingly be used in tool-assisted pipelines, for example to invoke spatial solvers or generate code, this work focuses on models' *direct, unaided* reasoning capabilities. FloorplanQA is designed to probe what LLMs can infer from structured input alone, without relying on external computation or visual grounding, in order to measure their unassisted capabilities. This baseline is important because even in tool-rich systems, models benefit from some unaided spatial ability to anticipate outputs and avoid trivial errors.

Specifically, our contributions are as follows:

- We introduce a dataset of 2,000 structured 2D layouts, including 600 each from synthetically generated kitchens, living rooms, and bedrooms, plus 200 layouts sourced from the Habitat Synthetic Scenes Dataset (HSSD) (Khanna et al., 2023), providing a realism check. All are represented in JSON and paired with spatial reasoning questions.

- We provide a diverse suite of 16,000 spatial reasoning questions, eight questions per layout, covering geometric relations, placement feasibility, spatial occupancy, and navigation.

- We establish structured evaluation protocols and scoring metrics that enable a fine-grained diagnosis of reasoning performance by task type and error mode.

- We conduct a comparative analysis of 15 LLMs, including 7 reasoning-focused models, as well as 8 standard models, revealing consistent failure patterns in spatial inference from symbolic input.

FloorplanQA provides a benchmark of layouts, questions, and evaluation metrics for assessing spatial reasoning in language models, focusing on symbolic floorplans that integrate geometry and semantics in ways that mirror real architectural abstractions.

## 2 RELATED WORK

Prior benchmarks have explored spatial reasoning across vision and language domains. CLEVR Johnson et al. (2017) is a synthetic visual question answering dataset designed to test compositional reasoning, including basic spatial relations. In real-world settings, SpatialSense Yang et al. (2019) focuses on recognizing spatial relations in images through adversarially mined examples. Benchmarks like BabyAI Chevalier-Boisvert et al. (2019), ALFRED Shridhar et al. (2020), and Room-to-Room (R2R) Anderson et al. (2018) integrate spatial understanding into embodied tasks, requiring agents to follow instructions involving navigation and object manipulation in simulated environments. Recent datasets such as ScanQA Azuma et al. (2022) and 3DSRBench Ma et al. (2024a) extend spatial reasoning evaluation into 3D environments, emphasizing the need for models to comprehend and reason about spatial relationships in three dimensions.

Vision-language models have advanced spatial reasoning but often handle it qualitatively. The VQA dataset Antol et al. (2015) challenges models to answer questions about images, while VL-T5 Cho et al. (2021) unifies vision-and-language tasks via text generation. Recent work on 3D scene graphs Armeni et al. (2019) introduces structured representations of environments, facilitating spatial reasoning. However, these approaches may miss fine-grained geometric details necessary for precise spatial inference. Efforts like SpatialVLM Chen et al. (2024) aim to endow vision-language models with enhanced spatial reasoning capabilities, addressing some of these limitations.

Advancements in generative models have also contributed to spatial reasoning tasks. Layout-GPT Feng et al. (2023) leverages large language models for compositional visual planning and layout generation, while Holodeck Yang et al. (2024) enables language-guided generation of 3D embodied AI environments. Similarly, AnyHome Fu et al. (2024) focuses on open-vocabulary generation of structured and textured 3D homes, highlighting the integration of language and spatial understanding in generative contexts. Infinigen Indoors (Raistrick et al., 2024) offers richly rendered 3D scenes but often produces implausible object placement due to non-convergent simulated annealing. LayoutVLM Sun et al. (2024) and FirePlace Huang et al. (2025) improve layout generation via optimization and constraint solving, respectively. But they assess output realism, not the model's ability to infer constraints directly. In contrast, our benchmark tests symbolic reasoning without tool-assisted refinement.

Evaluations of large language models' spatial understanding have been conducted in studies like Evaluating Spatial Understanding of Large Language Models Yamada et al. (2024), which assesses the spatial reasoning capabilities of LLMs through structured tasks. Additionally, benchmarks such as BALROG Paglieri et al. (2025) test agentic reasoning in game environments, further exploring the spatial and decision-making abilities of language and vision-language models. While these efforts reveal important limitations in high-level spatial understanding, our benchmark isolates low-level geometric reasoning in structured layouts, providing fine-grained and task-specific insights into models' spatial competence. Recent 3D-LLM surveys such as (Ma et al., 2024b) cover tasks like navigation and interaction, but not symbolic spatial reasoning. FloorplanQA fills this gap by testing raw spatial competence from structured layouts without multimodal input.

FloorplanQA addresses the gap in existing benchmarks by directly evaluating structured spatial inference from symbolic room layouts. Unlike prior benchmarks relying on raw images or focusing on commonsense spatial language, FloorplanQA provides explicit spatial representations (object

coordinates and dimensions) and tests models' abilities to perform precise spatial reasoning tasks, such as calculating distances, assessing visibility, and verifying object fit within a controlled setting.

# 3 METHOD

## 3.1 SYNTHETIC LAYOUT GENERATION

Our initial aim was to use publicly available real-world floorplan datasets. However, a comprehensive review revealed significant limitations, as several prominent datasets such as SUNCG (Song et al., 2017) and HouseExpo (Li et al., 2019) are not accessible due to unresolved copyright claims. Other large-scale resources—including 3D-FRONT (Fu et al., 2020), Structured3D (Zheng et al., 2020), and InteriorNet (Li et al., 2018)—are procedurally generated but impose constraints on layout diversity, furniture semantics, or downstream reuse. Datasets like CubiCasa5K (Kalervo et al., 2019) and Rent3D (Liu et al., 2015) offer fixed architectural plans from real environments but lack furnishing annotations. RPLAN (Wu et al., 2019), despite its scale, is not publicly released, and the dataset of Di et al. (2020), while large, is procedurally generated with realtor supervision and imposes restrictions on reuse. Given these legal, practical, and methodological constraints, we found synthetic data generation to be the most viable alternative.

We generated 1,800 synthetic indoor layouts using Gemini 2.5 Pro, a large language model fine-tuned for spatial reasoning (Google Gemini Robotics Team, 2025). Although our evaluation includes multiple LLMs (including Gemini variants), there is no circularity: the data-generation step is one-off and separate from evaluation. For evaluation, models solely produce answers to the benchmark questions, and correctness is computed by deterministic geometric routines against ground-truth solutions derived from the layouts, so no model output is used to generate data or to grade itself.

The generation process comprises two stages. First, we specify room geometries using explicit constraints on shape, adjacency, and design principles related to circulation, symmetry, and zoning. These constraints are encoded directly in the LLM prompt. Second, each room is furnished according to style-specific guidelines (e.g., a bedroom must contain a bed and storage), also defined in structured prompts, to encourage both visual realism and functional plausibility. Approximately one-third of candidate layouts are filtered out by a rule-based spatial validity filter that enforces basic clearance and accessibility constraints. The checks remove scenes with inaccessible furniture and implausible adjacencies, such as sofas blocking doors or a refrigerator overlapping a table; see Appendix B.3 for the full set of cases. Full prompts, generation templates, and validation scripts are provided in the Supplementary Material.

## 3.2 LAYOUT EXTRACTION FROM HSSD DATASET

To complement the synthetically generated layouts, we further incorporated 200 layouts extracted from the Habitat Synthetic Scenes Dataset (HSSD-200). HSSD provides 211 high-quality, human-authored 3D scenes designed with the Floorplanner interface and populated with 18,656 objects across 466 semantic categories. Unlike purely procedural datasets, HSSD offers fine-grained semantics, 3D assets, and close correspondence to real interiors, making it an effective proxy for real-world interior layouts.

For our purposes, we project each 3D scene to a 2D floorplan and retain only select structural and furniture elements. Decorative or auxiliary objects (e.g., *vases, plants, cushions, artworks, posters, bottles, shoes, candles*) are removed to reduce clutter; see Appendix C for the full details. We then use an $\alpha$-convex hull (Asaeedi et al., 2014; Edelsbrunner & Mücke, 1994) to smooth object boundaries, yielding polygonal layouts that are not restricted to axis-aligned rectangles. This step is necessary because raw HSSD projections often produce overly dense polygons, with redundant vertices along straight or nearly straight segments; applying an $\alpha$-hull reduces spurious complexity while preserving concavity, which avoids unnecessary token overhead in downstream LLM processing. This ensures compatibility with our synthetic layouts, while maintaining the richer geometric variety of HSSD.

Table 1: Left side: The illustration of the overall breakdown by room type for the entire 2,000-layout benchmark, encompassing both the synthetic component and the layouts extracted from the HSSD. Right side: Detailed distribution of the synthetic subset of the FloorplanQA benchmark (1,800 layouts) across room type, internal style, and geometric configuration.

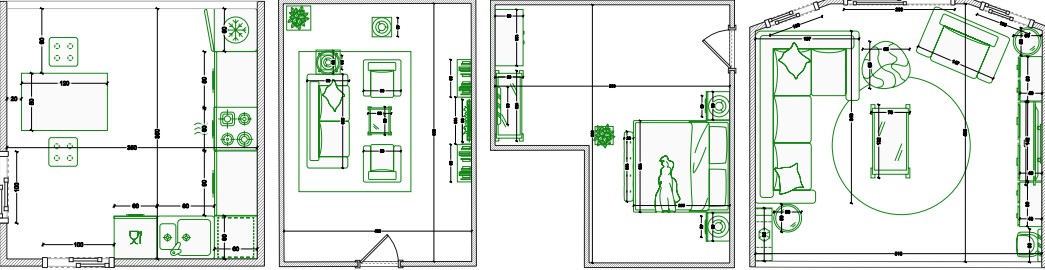

| Room Type | Style | Rectangular | L-Shaped | Open | Total |
|---|---|---|---|---|---|
| Kitchen | L-Shaped | 208 | 36 | 105 | |
| | U/G-Shaped | 42 | 128 | 2 | 600 |
| | Island-Based | 2 | 0 | 33 | |
| | Wall / Galley | 17 | 10 | 17 | |
| Living Room | Fireplace-Centric | 55 | 1 | 34 | |
| | Conversational | 66 | 3 | 37 | 600 |
| | Multi-Zone | 73 | 176 | 46 | |
| | TV-Focused | 78 | 3 | 28 | |
| Bedroom | With Workstation | 98 | 35 | 28 | |
| | Traditional | 114 | 59 | 57 | 600 |
| | Efficient Small | 65 | 3 | 8 | |
| | Welcoming Guest | 75 | 30 | 28 | |
| **Total** | | 893 | 484 | 423 | 1800 |

Figure 1: Representative layouts from FloorplanQA. **Generated**: kitchen, living room, bedroom. **HSSD**: living room (last image). Generated objects are axis-aligned boxes; HSSD uses arbitrary polygons.

## 3.3 UNIFIED DATASET

Together, these two sources yield a dataset of 2,000 layouts: 1,800 synthetically generated via Gemini 2.5 Pro and 200 extracted from HSSD. The two subsets share a unified polygonal representation, enabling consistent downstream processing. Figure 1 shows examples from both sources while Table 1 summarizes room, style, and geometry distributions across the synthetic subset with the figure next to it providing a breakdown by room type.

## 3.4 QUESTION TAXONOMY AND PROMPTING

FloorplanQA assesses spatial reasoning by presenting models with a single natural language question per symbolic layout. Questions span a range of topological and functional types, including numeric computations (distances, areas), spatial feasibility (object placement), visibility, and requirement violations. Some questions require fine-grained metric reasoning, others test whether a model can respect physical constraints. A categorized list of question types is shown in Table 2.

Each question is generated by filling a parameterized template with layout-specific variables such as object names, measurements, and task-specific contextual information (e.g. units for distance, clearance for paths, or which object-types should not occlude visibility). Prompts are issued in zero-shot settings, without few-shot examples or role-based instructions. Instead, we enforce simple structural markers—such as a required checklist and a final-answer line—to encourage stepwise reasoning.

To ensure verifiable outputs, each prompt specifies a response schema consisting of a brief structured justification and a final answer line. For example, a distance query is phrased as:

Table 2: FloorplanQA question taxonomy. The example question shown is an instantiation of the template used to generate all questions of that type. Each task is labeled with a format code: **N** (scalar), **B** (boolean), **S** (sequence), and **L** (list), and a question reasoning category.

| Type | Example Question | Format | Category |
|------|------------------|--------|----------|
| Distance | Calculate the Euclidean distance in meters between the centroids of the fridge and the stove | N | Metric |
| Free Space | Calculate the total non-occupied floor area in square meters | N | Topology |
| View Angle | Compute the smallest absolute angle in degrees between the vector from the centroid of the sofa to the centroid of the TV and the global north vector (0, 1) | N | Metric |
| Repositioning | Calculate how far the ottoman be moved in the left direction until it touches another object or the wall | N | Dynamic |
| Max Box | Calculate the area in square meters (m²) of the largest rectangle that can fit inside the room | N | Topology |
| Placement | Check if a 2m × 3m desk table can fit fully inside the room without overlaps | B | Topology |
| Shortest Path | Determine the shortest valid path that maintains a clearance of 15 cm from all other objects, starting from centroid of the stove and ending at the centroid of door | S | Dynamic |
| Visibility | Find all objects that intersect the vector from the centroid of the window to the centroid of the fireplace | L | Topology |

---

**Prompt: Distance Query**

```
Given the layout of a {room_type} in {format},
calculate the Euclidean distance in meters between the
centroids of `{obj1}` and `{obj2}`.
```

If the format, object names, or required inputs are missing, invalid, or inconsistent, the model must return: `*Final answer*: ERROR`. Otherwise, responses must follow the scheme, for example:

**Response Schema**

```
Begin by providing a concise checklist (3--7 bullets)
of the conceptual steps necessary for calculating the Euclidean
distance. Then, carefully walk through each reasoning step
required to calculate the distance.

Respond in the following strict format:
### Output Format
<step-by-step calculations>
*Final answer*: <answer>
```

This structure invokes step-by-step reasoning and each question ends with `*Final Answer*: <answer>`, enabling robust extraction even when APIs lack native structured-output support.

Layouts are represented in a structured JSON format. Each entry contains a `layout_id`, the `room_type`, and explicit geometric descriptions. The `room_boundary` is stored as a closed polygon, while `walls` are represented separately as a list of line segments. Openings such as windows and doors are included in a dedicated `openings` field rather than flattened into the object list. All furnishings and functional elements (e.g., bed, sofa, table) are stored in the `objects` list,

with each object defined by a labeled polygon. In the synthetic data, these polygons are axis-aligned bounding boxes (four points), whereas in HSSD they can exhibit arbitrary shapes and orientations. Object names are suffixed with instance identifiers (e.g., `fridge_1`, `table_3`) to ensure that referents remain unique and stable across prompt construction and answer evaluation. Coordinates are expressed in meters in a right-handed 2D Cartesian frame ($+x$ to the right, $+y$ upward). The global origin $(0, 0)$ is not fixed to the layout's lower-left corner and may vary across layouts. The prompt used to generate examples according to this schema is in Appendix I.

We group these questions into three reasoning categories. **Metric** tasks require explicit numerical computation, such as calculating centroids, measuring distances between objects, or evaluating the angle between an inter-object vector and a reference axis. **Topology** category involves geometric and relational reasoning, including checking placement feasibility, computing free space, or identifying whether an object blocks the direct line of sight between two others. **Dynamic** category addresses layout-changing procedures, such as repositioning an object until contact with a boundary or another object, or computing a valid collision-free path between two objects.

These categories are intended to capture the core modes of spatial reasoning in FloorplanQA, ranging from low-level geometric calculation to higher-level relational and procedural inference. While not strictly disjoint, they provide a diagnostic framework for analyzing model behavior and diagnosing failure modes.

In addition to categorizing by reasoning type, each task is also associated with an answer format code that specifies the expected output structure and the corresponding scoring rule. Scalar outputs (**N**) are scored by relative error with a default tolerance of 2%; for complex area-computation tasks (e.g., `Free Space`), the tolerance is relaxed to 5%. These tolerances are chosen to accommodate minor numerical instability in LLM outputs while remaining strict. Sensitivity studies supporting the constant choice, including tolerance sweeps showing that thresholds affect only absolute accuracies and not model rankings, are provided in Appendix G.1. Sequence outputs (**S**) are evaluated with a Fréchet threshold of 0.6 m, approximating minimal human clearance, and must be valid (collision-free, no overlaps). Threshold sweeps for path validation are reported in Appendix G.2. List outputs (**L**) are evaluated by set equality. Together, the categories and format codes define the taxonomy summarized in Table 2, which reports task coverage and examples for each case.

## 3.5 EVALUATION PROTOCOL AND SCORING

Each question in FloorplanQA is paired with a reference answer computed directly from the symbolic layout, enabling fully automated and deterministic evaluation of model outputs. Depending on the response type, correctness is assessed using numeric comparison with fixed tolerances, string matching, or geometric validation checks.

For numerical questions (e.g., distances, areas, angles), predictions are accepted if they fall within a relative error threshold. Sequence outputs are evaluated for both format and semantic correctness. Rather than requiring exact coordinate matches, we validate paths using geometric plausibility conditions of collision avoidance and sufficient proximity to a ground-truth trajectory. Deviation thresholds are set conservatively to tolerate minor geometric variations without crediting qualitatively wrong routes.

To differentiate reasoning failures from extraction or formatting issues, we apply a regex-based parsing pipeline, covering a fixed set of expected answer patterns (e.g., '`*Final answer*`' tokens in lower case or with surrounding symbols). If no answer is produced, or if the extracted content does not match a valid format, we count the response as an error (which is also considered incorrect). We provide a detailed breakdown of accuracy in Appendix E. In our evaluations, the proportion of invalidly formatted answers is below 1%.

We also explicitly track cases where no answer is returned due to truncation (API: `stop_reason = Token Limit`), a failure mode that disproportionately affects reasoning-heavy models. An aggregated per-model summary of truncation rates, invalid-format proportions, and parser sensitivity is reported in Table 8 (Appendix **??**).

To account for truncation, we report in Appendix J both the percentage of responses truncated by token limits and an adjusted accuracy computed only over valid (non-truncated) answers. These adjusted accuracies can be interpreted as an approximate upper bound on performance under our

prompting setup, since with larger token budgets models could in principle complete more solutions at similar quality. We therefore do not interpret non-truncated accuracy as a fully fair standalone score, as truncation occurs primarily on the most challenging questions and can artificially inflate performance.

### 3.6 MODEL INFERENCE SETUP

All models are queried using standard chat-based completion APIs, with prompts constructed as described above. We evaluate both large and mid-size models, including reasoning and standard variants. Large reasoning-oriented and standard models are allocated up to 12,288 tokens per completion, enabling them to process long layout descriptions and produce multi-step outputs. Mid-size models, including the GPT family variants optimized for speed and Qwen3-30B, are limited to 8,192 tokens. These budgets reflect our observations that larger models generate longer intermediate justifications and thus consume more tokens.

GPT-5 is evaluated under a distinct configuration: its reasoning intensity can't be disabled, so we run it with reasoning and verbosity set to "low," with a maximum output length of 4,096 tokens, while GPT-5-mini is run with reasoning and verbosity set to "medium," with a maximum output length of 8,192 tokens.

No model receives fine-tuning or prompt adaptation specific to FloorplanQA. All prompts are zero-shot, with the system message and output formatting constraints held fixed. For each model, we use the default inference configuration provided by its vendor, with temperature set to 0. The only exception is GPT-5, for which the temperature cannot be modified and defaults to 1.

Each model is evaluated over 1800 generated layouts and 200 layouts from HSSD, with one question from each type posed per layout. Evaluation is fully automated, from layout serialization and prompt insertion to parsing, with no manual curation.

All models are evaluated on identical input distributions and scoring criteria, enabling cross-system comparisons that are architecture-agnostic and directly comparable.

## 4 RESULTS

We evaluated the performance of the model on a dataset of 2,000 layouts, consisting of 600 kitchens, 600 living rooms, 600 bedrooms, and 200 additional layouts from HSSD. Each layout is paired with one question sampled from a pool of 8 parameterized templates, filtered by room applicability. This yields 16,000 layout–question pairs. The full taxonomy is shown in Table 2.

We evaluate fifteen models, spanning a wide range of parameter scales, architectures, and training regimes. The reasoning-oriented models include GPT-5 (OpenAI, 2025a), GPT-OSS-120B (OpenAI, 2025b), DeepSeek-R1-0528 (DeepSeek-AI, 2025), GPT-5-mini, Gemini Flash 2.5 (Google DeepMind, 2025), GPT-OSS-20B, and Qwen3-30B-A3B-Thinking-2507 (Yang et al., 2025). The general-purpose (standard) models include Claude Sonnet 4 (Anthropic, 2025), GPT-4.1 (OpenAI et al., 2024), Moonshot Kimi-K2-Instruct (Kimi Team, 2025), Qwen3-Coder-480B-A35B-Instruct, Qwen3-235B-A22B-Instruct-2507, GPT-4.1-mini, Qwen3-30B-A3B-Instruct-2507, and Devstral-Small-2505 (MistralAI, 2025). All models are evaluated in identical zero-shot conditions using standardized prompts and serialized layout inputs, as described in Section 3. Complete results disaggregated by question type, room type, and model are provided in Appendix E.

### 4.1 QUANTITATIVE RESULTS

We begin by aggregating accuracy across models and question types for both *reasoning* and *general* model families. Figure 2 summarizes accuracy: the top row shows general models, and the bottom row shows reasoning models. In each figure, the left panel summarizes accuracy by model, and the right panel summarizes accuracy by question across room types.

Kitchens lead across models because overlaps are rare, so most queries are straightforward. Scores on HSSD tend to lag behind the other room types; irregular, non-axis-aligned geometry and denser overlap make these layouts more demanding, see Appendix D for detailed statistics. Bedrooms and living rooms are mid-tier and nearly equal, lying between kitchens and HSSD.

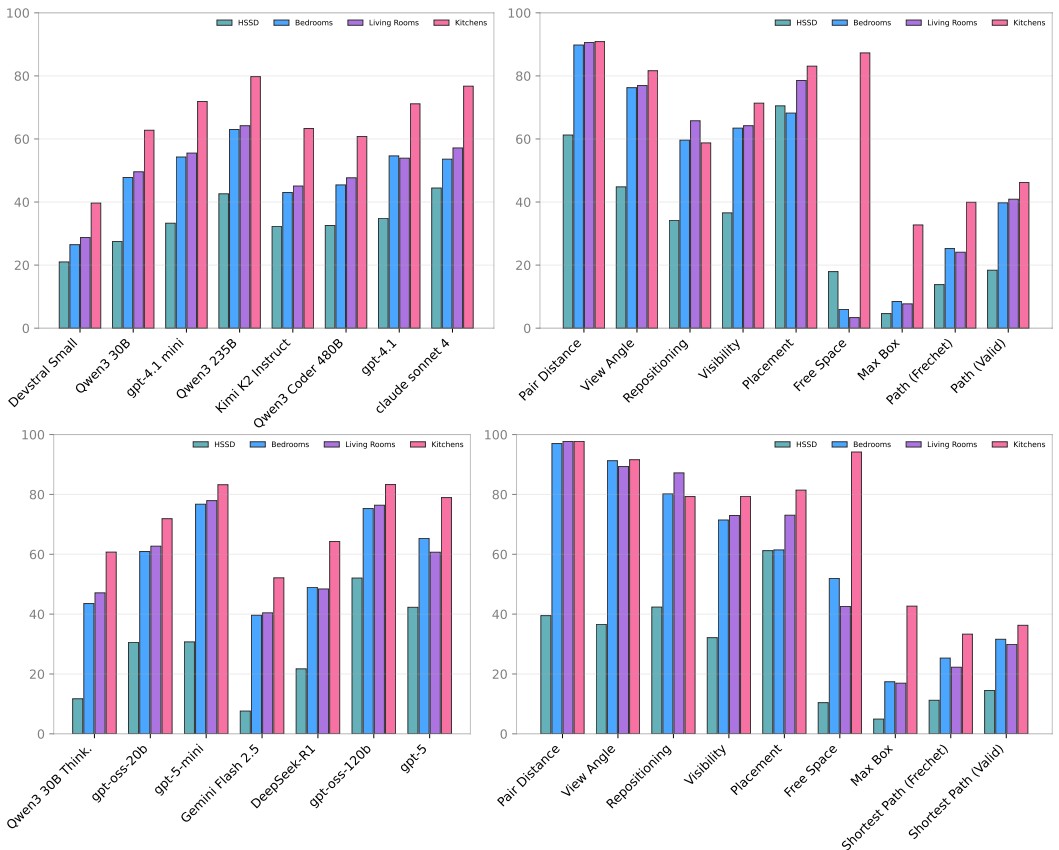

Figure 2: Accuracy of general (top) and reasoning (bottom) models. The left panel summarizes accuracy by model, averaged across all question types. The right panel summarizes accuracy by question, averaged across all models within the respective general or reasoning family. Each column corresponds to a specific room type represented in our dataset: **Kitchens**, **Living Rooms**, **Bedrooms** (synthetic subsets), plus **HSSD** layouts.

Comparing model families, general models struggle when many overlaps must be merged; they often treat object areas independently (no union), which hurts *Free Space* and *Max Box* and carries over into path planning. Reasoning models handle those cases with unions and rotations better, so they show gains on *Free Space* and *Max Box*. A practical limitation remains: **Gemini Flash 2.5** and **DeepSeek-R1** hit token limits on larger layouts, which drops scores, especially in HSSD.

Task difficulty follows a consistent pattern in terms of accuracy. Metric-category questions, such as *Pair Distance* and *View Angle*, achieve the highest accuracies. *Repositioning*, *Visibility*, and *Placement* yield mid-range accuracies. *Max Box* and *Free Space* benefit most from reasoning-oriented models, yet their accuracies remain low and are comparable in difficulty to *Shortest Path*. Accuracy decreases as object count and overlap density increase, most notably on HSSD layouts. A detailed analysis of each question category, with visualizations and representative failure cases, is provided in Appendix H.

## 4.2 ABLATION

To evaluate the robustness of layout interpretation under alternate encodings, we perform a **format ablation** that replaces the standard JSON layout with a semantically equivalent XML version. This substitution preserves all geometric and object-level content while modifying only the syntax and structural serialization. The **prompt level** and **semantic** ablations are described in Appendix, Sections G.3, G.4. To avoid recomputing the full suite, we focus on the most variance-sensitive question types for HSSD layouts, as shown in Appendix E.4 (Figs. 5 and 6): `View Angle`, `Visibility`,

Table 3: Accuracy using JSON vs XML layout encoding. Each cell shows performance on the original JSON representation and its equivalent XML rendering.

| Model | Repositioning | View Angle | Visibility |
|-------|--------------|-----------|-----------|
| GPT-OSS-120B | $60.5 \rightarrow 59.0$ | $74.0 \rightarrow 74.0$ | $70.0 \rightarrow 72.0$ |
| GPT-OSS-20B | $40.0 \rightarrow 39.0$ | $37.5 \rightarrow 38.5$ | $45.5 \rightarrow 43.5$ |
| Qwen3-235B-A22B | $39.0 \rightarrow 42.0$ | $50.5 \rightarrow 54.0$ | $70.5 \rightarrow 65.5$ |

and one task from a different category—`Repositioning (Dynamic)`. We apply each input-format ablation to three representative models: two reasoning models (GPT-OSS-120B, GPT-OSS-20B) and one large general model (Qwen3-235B-A22B) for the same subset of HSSD layouts. As shown in Table 3, accuracy is largely stable in both formats for most categories; the GPT-OSS models change minimally, while Qwen shows modest fluctuations. This suggests that these models encode layout semantics in a manner that is relatively invariant to low-level representation details.

Additionally, beyond input-format robustness, we evaluate whether external computation and visual renderings change performance. Tool-augmented (Python Code Interpreter) and VLM-based settings are analyzed in Section F: tools substantially improve scalar tasks, while gains are limited on dynamic planning tasks, and VLM inputs yield selective improvements without changing overall model trends.

## 5 CONCLUSION

We introduced **FloorplanQA**, a benchmark for spatial reasoning over symbolic 2D layouts aligned with architecture and robotics practice. We evaluated 15 language models (8 general, 7 reasoning) on 2,000 layouts (1,800 synthetic; 200 semi–real HSSD) across tasks ranging from metric queries to visibility, placement, and shortest path with clearance.

Empirically, metric and simple visibility queries are reliable; kitchens score highest because overlaps are rare. HSSD layouts are more demanding—irregular, non–axis-aligned shapes and dense overlaps expose weaknesses such as centroid miscalculation and missed unions. Reasoning models improve notably on *Free Space* and *Max Box* by handling overlaps and rotations more consistently, while general models often subtract object areas independently and fail under heavy overlap. *Shortest Path* is sufficiently challenging because it requires multiple correct steps (clearance buffering, collision checks, and path search), where errors compound.

These results indicate that current LLMs lack sufficiently robust internal geometric representations for complex spatial inference. We also ran a small set of capability extensions to probe where models benefit from extra modalities. Enabling a Python Code Interpreter yields strong gains on arithmetic-heavy scalar tasks (distance, angles, visibility), while harder optimization and planning tasks (e.g., `Max Box` and `Shortest Path`) remain challenging because failures often stem from incorrect spatial reasoning or imperfect model-written code. Providing rendered floorplan images to VLMs yields improvements on some visually grounded cases (such as object fit and certain metric cues), but does not consistently increase overall performance across tasks, indicating that symbolic input already provides a strong baseline and that visual benefits depend on ability to understand the rendering representation.

Two complementary directions follow. *Near term*: hybridize with external geometric solvers or symbolic planning modules—set operations (unions/differences), centroid via shoelace, clearance buffering, oriented rectangle search, and A* path planning—to compensate for the models' weaknesses in collision avoidance and clearance reasoning. *Longer term*: train with explicit spatial constraints and harder distributions (irregular, overlap–heavy layouts), and include constraint–violation exemplars and geometry-aware objectives so models learn to maintain coherence under rotation, clearance, and union operations in design-oriented tasks. Beyond these preliminary directions, potential improvements include multi-step interaction (e.g., asking an agent to verify or revise its solution using a rendered floorplan). We view these as promising follow-up projects, while FloorplanQA already provides a strong standalone benchmark to measure progress in spatial reasoning for layout design.

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

## A  TECHNICAL APPENDIX AND SUPPLEMENTARY MATERIAL FOR FLOORPLANQA

This appendix details the dataset generation pipeline, dataset statistics, and the full evaluation of *FloorplanQA* across reasoning and general models. It also includes ablation studies, analysis of failure cases, and the exact prompts used for data generation and question asking.

All artifacts (code, generation prompts, evaluation scripts, and visualization utilities) are included with the submission as supplementary material.

## B  DETAILS ON SYNTHETIC DATA GENERATION

Room layouts were generated using Gemini 2.5 Pro, specifically fine-tuned for spatial reasoning and bounding-box tasks. In the following sections, we describe the detailed procedure and its constraints. All generation scripts, prompts, constraint-checking code, and random seeds are included in the code for full reproducibility.

### B.1  ROOM SHAPE GENERATION

We generated 600 layouts for each of the kitchens, bedrooms, and living rooms. These layouts feature a range of geometries and sizes, with clearly defined room structures that incorporate windows, doors, and corner cutouts where applicable. All layouts follow specifications tailored to each room type.

Each layout falls into one of three shape categories: rectangular, L-shaped, or open. Size categories were defined individually for each room type and were incorporated into the generation prompts based on standard assumptions about typical room dimensions.

The distribution of room shapes and sizes is shown in Table 4, and example layout types are illustrated in Fig. 1.

Table 4: Shape and size distribution across generated layouts for each room type.

| Room Type | Shape Distribution (Rect / L-shaped / Open) | Size Categories (in m$^2$) (Small / Medium / Large) |
|---|---|---|
| Kitchen | 40% / 40% / 20% | $\leq 7$ / 7–18 / >18 |
| Bedroom | 50% / 30% / 20% | 8–12 / 12–18 / >18 |
| Living Room | 40% / 40% / 20% | 20 / 22 / 24 |

**Geometric and Structural Constraints:**  The geometry of the room was procedurally varied using prompt-based guidance to produce rectangular, L-shaped, and open-plan configurations. The following structural properties were described in the prompts but not explicitly enforced during layout generation:

- L-shaped rooms were described as rectangular spaces with a square cutout in one corner. To maintain usable proportions, each leg of the L shape was suggested to be at least 1.5 m wide and deep.

- Open-plan rooms, apart from living rooms, were prompted without doors and with one whole wall removed. This was intended to vary the room's shape and simplify the layout generation process.

- Doors were described with widths between 0.8 m and 1.0 m, randomly selected. The windows were chosen from a fixed set of widths: 0.6 m, 0.75 m, 0.9 m, 1.2 m, and 1.5 m.

- The prompts included instructions on placing all elements, such as doors, windows, and cutouts, entirely within the room boundaries.

**Window Placement:** Window placement followed prompt-based guidelines aimed at supporting daylight access and layout clarity:

- The total window length was set to exceed 15 percent of the room's floor area. It was used as a simple proxy to ensure visible window openings and visual balance along the walls.

- The windows on the same wall were the same size to support visual balance. Small, isolated windows were not used.

- Long windows, over 1.5 m, were split into segments with 0.05 to 0.15 m gaps for a more modular appearance.

- The windows were not placed opposite each other or on the same wall as a door, as such arrangements are less common and were not emphasized in the prompt design.

### B.2    Furniture and Appliance Placement

In the second stage, each room was populated with furniture and appliances based on layout style and object-specific constraints. While the styles differ by room type–such as enforcing a work triangle in kitchens, orienting seating around a focal point in living rooms, or centering beds symmetrically in bedrooms–the overall placement process followed a unified set of rules:

- All floor-standing objects must be placed without overlaps, except in semantically grouped cases (e.g., lamps on tables, chairs under tables).

- Clearance zones must be preserved around doors, main pathways, and functional elements such as beds, appliances, and desks.

- Placement follows a priority order: essential furniture is placed first, followed by optional and decorative elements only if space allows.

- Major objects like fridges, ovens, and beds must be anchored to structural walls or room boundaries.

Layouts violating any of these hard constraints, due to overlap, clearance issues, or improper attachment, were automatically discarded.

### B.3    Layout Selection Criteria

Approximately one-third of the initially generated room layouts were filtered out using a set of geometric and functional constraints. These filters were designed to ensure realistic object placement, functional usability, and architectural plausibility. While these rules are based on general design principles, they are not based on any specific design standard. Instead, they are the result of iterative development, focusing specifically on addressing cases where our prompts lead to unlikely or implausible layouts. After filtering, we retained 600 valid layouts for each room type.

- **Non-overlapping objects (with exceptions)**: Each pair of objects must satisfy axis-aligned bounding box (AABB) separation constraints, unless they belong to a known exception category. For two objects $A$ and $B$ with bounding boxes $(x_1^A, y_1^A, x_2^A, y_2^A)$ and $(x_1^B, y_1^B, x_2^B, y_2^B)$, non-overlapping requires:

$$x_2^A \leq x_1^B \quad \text{or} \quad x_2^B \leq x_1^A \quad \text{or} \quad y_2^A \leq y_1^B \quad \text{or} \quad y_2^B \leq y_1^A$$

  This constraint is not enforced for the following semantically compatible object pairs: (i) `rug` with any object placed on top of it; (ii) `lamp` with `nightstand`, `desk`, or `table`; (iii) `tv` with `tv_stand`; (iv) `chair` objects with `desk` or `table`.
  These exception pairs are considered contextually collocated or hierarchically related (e.g., support/surface relationships) and are therefore allowed to overlap.

- **Non-blocking door clearance**: Doors have physical thickness and are defined by bounding boxes $(x_1^d, y_1^d, x_2^d, y_2^d)$. A clearance zone of $door\_length$ meters is required in front of the door to ensure swing space and accessibility. The position of this zone depends on which wall the door is attached to.

- **No windows on opposite walls**: We did not include layouts with windows on directly opposite walls, as such configurations are uncommon in typical residential designs.
- **Appliances against walls or cutout edges**: Large fixtures like fridges and ovens must be flush against at least one wall or cutout boundary. This is formalized by enforcing:

$$x_1 = 0 \quad \text{or} \quad x_2 = W \quad \text{or} \quad y_1 = 0 \quad \text{or} \quad y_2 = D \quad \text{or edge of cutout}$$

For rooms with cutouts, an object may align with a cutout boundary, defined as additional wall segments with known coordinates.

These constraints were iteratively selected to address common implausible layouts generated by Gemini 2.5 Pro using our prompts. They are not universal requirements for real layouts, nor a complete set of constraints, but aim to avoid frequent sources of implausibility in generated layouts.

## C   DETAILS ON HSSD LAYOUTS SELECTION

We curate a subset of HSSD layouts to ensure clean geometry and unambiguous supervision for spatial reasoning tasks. Starting from the raw scenes, we generate a 2D floor-plan projection and apply filtering and normalization steps (e.g., geometry cleanup and category unification).

**Object filtering.**   To reduce visual clutter and retain only objects essential for spatial reasoning, we exclude purely decorative or small accessory categories. The banned labels are: `accessory`, `air conditioner`, `artwork`, `blanket`, `boots`, `bottle`, `bowl`, `box`, `book`, `brush`, `candle`, `cushion`, `decor`, `dog`, `fan`, `flower`, `frame`, `guitar`, `herb`, `hook`, `jar`, `light`, `lightbulb`, `orchids`, `pendant`, `pendulum`, `pet`, some `plants`, `poster`, `pot`, `shoes`, `socket`, `switch`, `vase`, `wreath`. We also drop subcomponents that fragment footprints without changing free space (e.g., chair/table legs; bed/armchair frames), remove partial cabinet-door leaves, and consolidate multiple near-duplicate small items by keeping a single representative instance. The resulting scenes preserve the functional layout while simplifying geometry. In Figure 3, the left panel shows the layout immediately after 2D projection; the right panel shows the simplified layout after filtering and normalization.

**Projection and cleanup.**   During 2D projection, we correct mislabeled *windows* and *doors*, snap nearly collinear edges, and resolve self-intersections. To smooth rectilinear artifacts and irregular edges in box-like footprints, we apply *alpha-shape*–based convex hulls, then recompute centroids and areas on the cleaned geometry.

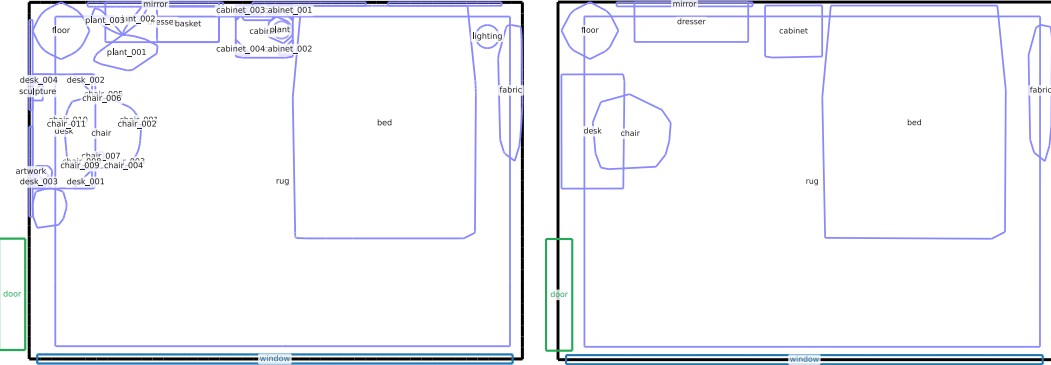

Figure 3: HSSD layout selection: comparison of the original (left) and simplified (right) 2D layouts. The simplified version removes redundant or overlapping objects and cleans geometry to produce well-structured input for spatial reasoning tasks.

## D   ROOM STATISTICS

Table 5 reports summary statistics for the layouts. Kitchens are typically the smallest spaces, with relatively few objects and overlaps, but a high density due to their compact geometry. Living rooms

are the largest, with slightly more objects overall but lower density, reflecting their open layout. Bedrooms fall in between, with similar object counts to kitchens but more frequent overlaps.

The HSSD layouts are comparable in scale to bedrooms and living rooms in terms of area and object count, but differ in structure: objects are represented with detailed, non-axis-aligned polygons, resulting in a significantly higher vertex count. They also exhibit more overlaps than the generated layouts, reflecting their closer alignment with human-authored floorplans. In other respects, however, the distributions remain broadly consistent.

Table 5: Average layout statistics by room type.

| Metric | Kitchen | Bedroom | Living Room | HSSD |
|---|---|---|---|---|
| Avg. Area (m$^2$) | 12.00 | 17.76 | 20.75 | 17.95 |
| Avg. # of Objects | 10.35 | 10.76 | 11.69 | 12.20 |
| Avg. # of Overlaps | 0.52 | 1.82 | 1.52 | 4.39 |
| Avg. Object Density | 0.95 | 0.66 | 0.57 | 0.83 |
| Avg. Vertices per Layout | 41.39 | 43.03 | 46.77 | 152.29 |

To further illustrate these statistics, Figure 4 shows the distribution of object counts across all layouts. The histograms confirm the averages reported in Table 5: kitchens are concentrated at lower counts, typically around 10 objects; bedrooms and living rooms exhibit broader distributions with slightly higher counts; and HSSD layouts overlap in range but extend to higher counts in the tail. Overall, the object distributions remain comparable across sources, with no extreme outliers.

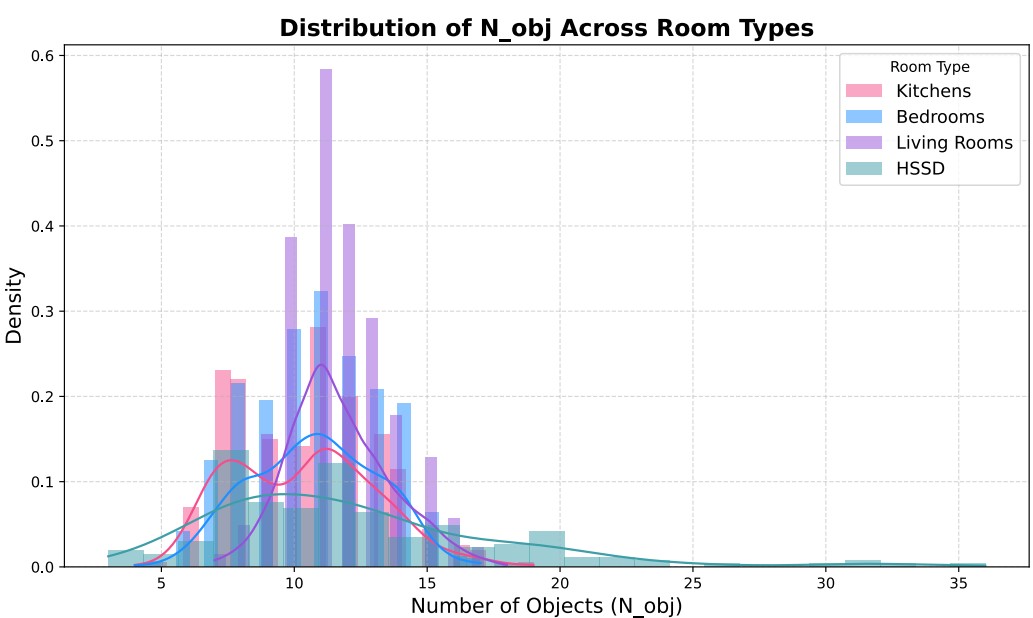

Figure 4: Distribution of object counts across kitchens, bedrooms, living rooms, and HSSD layouts.

# E  FULL BREAKDOWN BY ROOM TYPE, QUESTION TYPE, AND MODEL

## E.1  DETAILED ACCURACY BY FULL DATASET

Tables 6 and 7 report the full per-model accuracy matrix for the base and reasoning model groups, respectively. Each table covers nine question categories across four room types, resulting in a total of 36 rows. The `path` task is assessed using two complementary metrics: validity and Fréchet Distance. Together, these question types span a range of challenges, covering both reasoning-heavy

and functionally grounded tasks. The base group includes eight general-purpose models, while the reasoning group includes seven models with explicit reasoning capabilities. For every model, question type accuracy is computed over a fixed set of 600 synthetic layouts (Kitchens, Living Rooms, Bedrooms) and 200 HSSD layouts, ensuring that results are directly comparable across models.

Table 6: Question-level accuracy on full dataset by room type for **standard models**.

| Question | Room | claude sonnet-4 | gpt-4.1 | Kimi-K2 Instruct | Qwen3 Coder-480B | Qwen3 235B | gpt-4.1 mini | Qwen3 30B | Devstral Small |
|---|---|---|---|---|---|---|---|---|---|
| Pair distance | K | 99.8 | 96.5 | 95.7 | 96.8 | 99.2 | 90.7 | 89.5 | 58.8 |
| | LR | 99.5 | 95.0 | 94.2 | 96.8 | 99.7 | 88.5 | 88.8 | 62.3 |
| | B | 99.7 | 96.3 | 93.3 | 96.5 | 99.5 | 87.8 | 85.2 | 60.0 |
| | HSSD | 88.0 | 56.0 | 75.5 | 66.0 | 67.0 | 37.0 | 44.5 | 56.0 |
| Placement | K | 87.8 | 78.0 | 82.2 | 80.3 | 90.2 | 86.5 | 85.5 | 74.2 |
| | LR | 80.5 | 69.0 | 73.8 | 83.2 | 89.2 | 75.2 | 85.7 | 71.8 |
| | B | 68.8 | 59.8 | 67.5 | 70.8 | 76.7 | 68.7 | 77.0 | 56.3 |
| | HSSD | 72.0 | 64.5 | 73.0 | 70.0 | 82.0 | 76.5 | 71.5 | 54.5 |
| Reposi-tioning | K | 73.8 | 63.8 | 48.7 | 45.3 | 83.5 | 66.3 | 73.7 | 14.8 |
| | LR | 79.3 | 60.3 | 56.7 | 64.5 | 91.3 | 76.8 | 72.2 | 25.0 |
| | B | 71.0 | 55.5 | 48.3 | 59.5 | 79.0 | 72.5 | 70.0 | 21.2 |
| | HSSD | 42.0 | 47.0 | 28.0 | 34.0 | 39.0 | 40.5 | 33.0 | 10.0 |
| Free space | K | 97.8 | 93.2 | 83.0 | 84.2 | 95.0 | 95.2 | 83.8 | 66.2 |
| | LR | 0.2 | 14.2 | 2.8 | 1.8 | 3.5 | 1.2 | 0.5 | 2.7 |
| | B | 2.7 | 31.2 | 1.0 | 1.3 | 8.8 | 0.8 | 1.0 | 0.7 |
| | HSSD | 35.0 | 16.0 | 24.0 | 22.0 | 17.0 | 15.5 | 7.5 | 6.5 |
| Visibility | K | 87.7 | 63.2 | 54.5 | 67.0 | 98.3 | 90.2 | 91.5 | 18.5 |
| | LR | 81.7 | 52.7 | 43.3 | 52.2 | 98.3 | 86.8 | 88.7 | 10.0 |
| | B | 74.8 | 57.5 | 41.0 | 54.0 | 96.7 | 86.3 | 86.3 | 10.8 |
| | HSSD | 46.5 | 20.0 | 22.5 | 26.5 | 70.5 | 52.0 | 45.5 | 9.0 |
| View angle | K | 92.0 | 95.3 | 69.8 | 78.8 | 97.0 | 95.8 | 74.7 | 49.7 |
| | LR | 87.7 | 93.2 | 59.7 | 75.3 | 94.0 | 93.0 | 72.2 | 40.5 |
| | B | 88.0 | 90.2 | 60.3 | 72.8 | 95.0 | 91.2 | 76.8 | 35.8 |
| | HSSD | 67.5 | 55.0 | 28.5 | 46.5 | 50.5 | 46.0 | 34.5 | 30.0 |
| Max box | K | 47.2 | 31.8 | 32.0 | 26.8 | 65.5 | 27.5 | 26.5 | 4.5 |
| | LR | 7.8 | 7.0 | 5.0 | 5.7 | 22.3 | 4.5 | 8.8 | 0.5 |
| | B | 5.8 | 6.8 | 7.3 | 5.0 | 29.2 | 4.8 | 7.0 | 1.7 |
| | HSSD | 5.0 | 7.5 | 2.5 | 2.0 | 11.5 | 4.5 | 3.0 | 1.0 |
| Shortest path (valid) | K | 59.2 | 61.7 | 52.7 | 39.7 | 45.2 | 55.3 | 21.8 | 34.2 |
| | LR | 53.0 | 51.3 | 42.8 | 37.3 | 44.8 | 47.2 | 20.3 | 30.3 |
| | B | 48.5 | 52.2 | 40.7 | 34.7 | 44.2 | 45.8 | 18.5 | 33.5 |
| | HSSD | 28.5 | 25.0 | 18.5 | 18.0 | 26.0 | 15.0 | 5.5 | 10.5 |
| Shortest path (Fréchet) | K | 45.3 | 56.8 | 51.3 | 28.2 | 44.2 | 39.5 | 17.8 | 36.2 |
| | LR | 24.7 | 42.5 | 27.3 | 12.3 | 34.7 | 26.5 | 9.2 | 15.5 |
| | B | 23.0 | 42.2 | 27.7 | 14.2 | 38.0 | 30.5 | 8.2 | 18.3 |
| | HSSD | 15.5 | 22.5 | 18.0 | 8.0 | 20.0 | 12.5 | 2.5 | 11.5 |

## E.2 Token-Limit Analysis and Valid-Only Accuracy

In addition to overall accuracy, we analyze two complementary aspects of model performance.

First, Tables 12 and 14 quantify the fraction of responses that were terminated due to the TOKEN LIMIT stop reason. This failure mode is particularly relevant for reasoning-oriented models, which often generate longer chain-of-thought outputs.

Second, Tables 13 and 15 report accuracy over *completed* answers only, excluding truncated outputs (e.g., token-limit terminations). This metric isolates models' reasoning performance on successfully produced, well-formed responses.

Together, these analyses complement the full accuracy tables by disentangling reasoning failures from generation truncation and formatting issues.

## E.3 E.3 Aggregated Truncation Rates

Table 8 reports aggregated truncation and error-type statistics for each model, computed by averaging outcomes across all room types and all question categories in the full dataset. A response

Table 7: Question-level accuracy on full dataset by room type for **reasoning models**.

| Question | Room | gpt-5 | gpt-oss 120b | DeepSeek R1-0528 | Gemini Flash 2.5 | gpt-5 mini-2025 | gpt-oss 20b | Qwen3 30B Think. |
|---|---|---|---|---|---|---|---|---|
| Pair distance | K | 99.8 | 99.3 | 98.0 | 96.3 | 100.0 | 94.2 | 97.7 |
| | LR | 98.8 | 99.3 | 99.0 | 96.0 | 99.7 | 93.5 | 97.5 |
| | B | 98.3 | 99.5 | 96.8 | 95.5 | 99.7 | 93.8 | 95.3 |
| | HSSD | 69.0 | 78.5 | 25.5 | 12.5 | 32.5 | 40.5 | 18.0 |
| Placement | K | 84.7 | 92.0 | 89.0 | 59.7 | 90.8 | 85.7 | 68.2 |
| | LR | 75.5 | 89.0 | 82.2 | 53.3 | 86.3 | 78.5 | 46.5 |
| | B | 61.2 | 83.5 | 72.0 | 35.8 | 81.2 | 62.0 | 34.5 |
| | HSSD | 70.0 | 85.0 | 79.0 | 16.0 | 75.5 | 74.5 | 28.5 |
| Reposi-tioning | K | 83.0 | 85.5 | 79.2 | 90.5 | 84.5 | 70.8 | 61.5 |
| | LR | 85.5 | 89.8 | 86.2 | 91.2 | 92.8 | 87.8 | 77.0 |
| | B | 77.8 | 83.3 | 83.3 | 85.2 | 84.3 | 78.0 | 69.2 |
| | HSSD | 49.5 | 60.5 | 47.5 | 18.5 | 53.5 | 40.0 | 27.0 |
| Free Space | K | 82.5 | 99.0 | 93.0 | 93.3 | 99.5 | 94.8 | 97.0 |
| | LR | 47.0 | 83.3 | 18.3 | 17.7 | 78.5 | 53.2 | 0.0 |
| | B | 50.5 | 87.5 | 34.8 | 33.3 | 82.2 | 74.0 | 1.2 |
| | HSSD | 19.5 | 31.0 | 6.5 | 1.0 | 5.0 | 9.0 | 1.0 |
| Visibility | K | 94.8 | 94.2 | 71.3 | 26.8 | 98.0 | 91.5 | 78.8 |
| | LR | 95.2 | 94.0 | 52.0 | 11.3 | 98.0 | 89.3 | 70.8 |
| | B | 94.2 | 92.5 | 53.5 | 11.2 | 95.5 | 89.2 | 64.2 |
| | HSSD | 57.0 | 70.0 | 10.0 | 0.5 | 39.0 | 45.5 | 3.0 |
| View Angle | K | 96.2 | 98.5 | 73.7 | 92.5 | 88.3 | 93.5 | 98.5 |
| | LR | 93.3 | 97.3 | 68.2 | 91.5 | 84.5 | 92.0 | 98.3 |
| | B | 95.2 | 98.2 | 75.2 | 93.8 | 86.8 | 91.3 | 98.3 |
| | HSSD | 59.5 | 74.0 | 13.5 | 20.0 | 25.5 | 37.5 | 26.0 |
| Max Box | K | 48.5 | 62.8 | 50.5 | 3.7 | 85.2 | 31.3 | 16.7 |
| | LR | 17.3 | 28.2 | 8.0 | 0.0 | 60.3 | 3.8 | 0.8 |
| | B | 13.0 | 30.3 | 11.0 | 0.0 | 61.2 | 5.8 | 0.5 |
| | HSSD | 5.0 | 9.5 | 2.5 | 0.0 | 17.0 | 0.5 | 0.0 |
| Shortest path (valid) | K | 64.7 | 64.2 | 12.3 | 3.2 | 52.3 | 43.0 | 14.2 |
| | LR | 21.2 | 66.0 | 12.5 | 1.5 | 53.7 | 37.7 | 16.7 |
| | B | 58.2 | 57.8 | 7.8 | 1.0 | 52.0 | 30.0 | 14.3 |
| | HSSD | 28.5 | 33.5 | 8.5 | 0.0 | 16.5 | 13.5 | 1.0 |
| Shortest path (Fréchet) | K | 56.3 | 55.5 | 11.5 | 3.2 | 50.5 | 42.2 | 14.0 |
| | LR | 12.3 | 40.5 | 9.3 | 1.3 | 47.5 | 28.5 | 16.3 |
| | B | 39.3 | 44.8 | 6.0 | 0.8 | 47.7 | 24.2 | 14.3 |
| | HSSD | 22.5 | 26.5 | 2.5 | 0.0 | 12.0 | 14.0 | 1.0 |

is counted as **truncated** when it terminates with the TOKEN LIMIT stop reason. We additionally report the fraction of **invalid-format** responses that could not be parsed into a valid answer under our evaluation protocol.

For completeness, Table 8 also includes the proportions of **wrong** and **correct** responses, as well as an **alternative accuracy** (% alt), computed by extracting the last numeric value or the final list from each model's output. This alternative metric serves purely as a robustness check on the parsing and evaluation pipeline.

### E.4 RADAR SUMMARIES

For readability, we also provide radar visualizations that summarize accuracy and variance across models and question types, complementing the main tables.

**General models.** Figure 5 shows mean accuracy (top left) and variability (top right) across general models, as well as accuracy (bottom left) and variability (bottom right) by question type. Kitchens are consistently easier, while HSSD is the most challenging. Among models, Devstral Small performs noticeably worse, whereas Qwen3-30B achieves a level comparable to that of larger models. Across question types, Pair Distance and View Angle from the Metric category yield the highest accuracy, while more complex tasks such as Max Box and Shortest Path show lower scores and higher variance across rooms.

**Reasoning models.** Figure 6 presents analogous plots for reasoning models. GPT-family models show stronger results overall. In contrast, DeepSeek-R1 and Gemini Flash 2.5 struggle with token limits, as these models tend to produce very long outputs according to Table 14. By question type,

Table 8: Error-type distribution and accuracy on the full dataset. Values are aggregated for each model by averaging across all room types and all questions. **Truncation** (% trunc.) quantifies the fraction of responses that were terminated due to the `TOKEN LIMIT` stop reason. **Invalid-format** (% invalid) denotes responses that could not be parsed into a valid answer. **Alternative accuracy** (% alt) reports accuracy computed using the last numeric value or the last list in the model's output.

| Model | % trunc. | % invalid | % wrong | % correct | % alt |
|---|---|---|---|---|---|
| gpt-5 | 3.86 | 0.27 | 30.42 | 65.45 | 65.45 |
| gpt-oss-120b | 2.05 | 1.05 | 21.41 | 75.49 | 75.53 |
| DeepSeek-R1 | 31.31 | 0.78 | 17.26 | 50.65 | 50.65 |
| gpt-5-mini | 15.50 | 1.34 | 12.54 | 70.62 | 70.62 |
| Gemini Flash 2.5 | 50.58 | 0.13 | 10.14 | 39.15 | 39.15 |
| gpt-oss-20b | 16.94 | 1.18 | 21.70 | 60.18 | 60.31 |
| Qwen3-30B Think | 38.95 | 0.09 | 16.52 | 44.44 | 44.44 |
| claude-sonnet-4 | 0.05 | 0.07 | 41.09 | 58.80 | 58.80 |
| gpt-4.1 | 0.14 | 0.09 | 41.52 | 58.25 | 58.25 |
| Kimi-K2 | 0.15 | 0.03 | 52.04 | 47.78 | 47.78 |
| Qwen Coder-480B | 0.02 | 0.09 | 49.40 | 50.48 | 50.48 |
| Qwen-235B | 13.64 | 0.13 | 20.31 | 65.92 | 65.98 |
| gpt-4.1-mini | 0.67 | 0.10 | 42.18 | 57.05 | 57.05 |
| Qwen-30B Instr | 6.85 | 3.09 | 38.47 | 51.59 | 51.84 |
| Devstral Small | 2.75 | 1.56 | 65.64 | 30.05 | 30.07 |

`Repositioning` and `Placement` are handled reliably, whereas `Max Box` and `Shortest Path` remain the most difficult as well, with high variance across rooms.

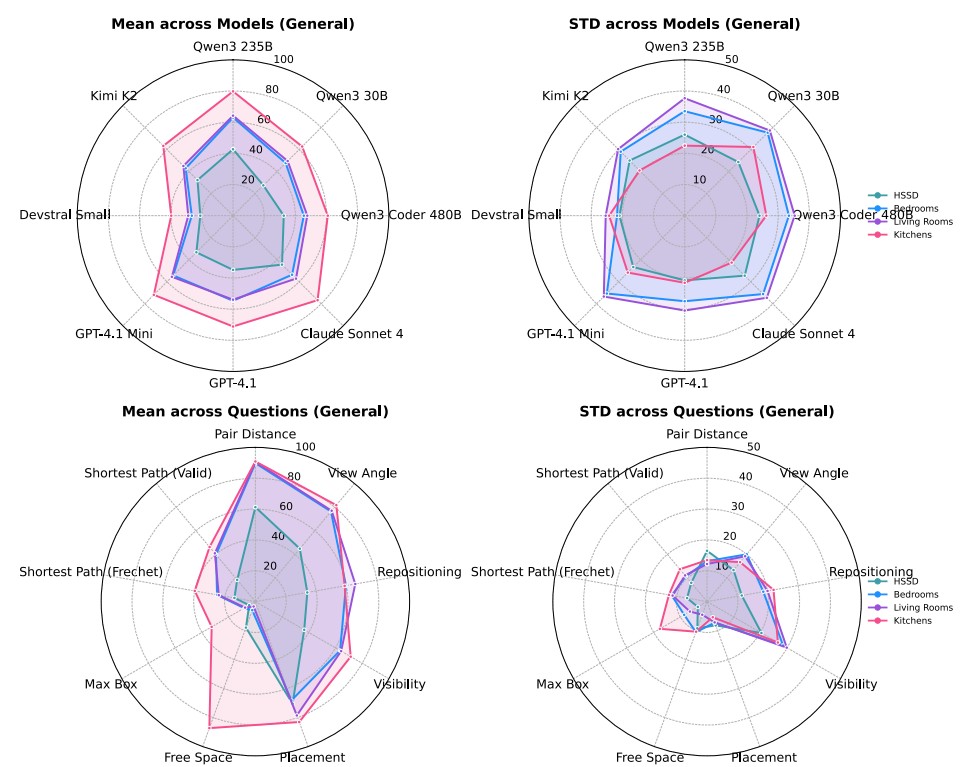

Figure 5: Radar plots for **general models**, showing mean and standard deviation of accuracy across (top) models and (bottom) question type.

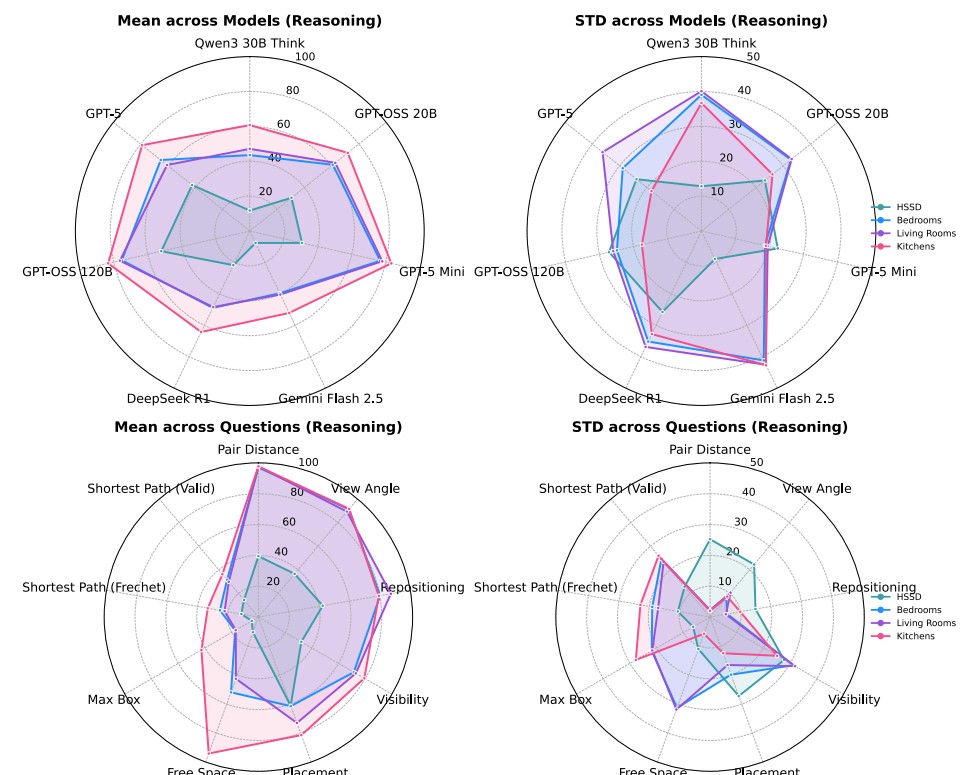

Figure 6: Radar plots for **reasoning models**, showing mean and standard deviation of accuracy across (top) models and (bottom) question type.

## F  TOOLS AND VLM EXPERIMENTS

To complement the text-only benchmark, we additionally evaluate (i) tool-augmented reasoning using an integrated Python interpreter and (ii) vision-language models (VLMs) using rendered floorplan images. These studies measure the potential benefit of external computation and visual input while keeping the task definitions and scoring identical to the main benchmark.

**Tool-augmented setting.**  We enable the Code Interpreter tool for GPT-4.1 and GPT-4.1-mini, allowing the model to invoke a Python sandbox during inference. The model is instructed to use Python whenever numeric computation is needed (e.g., distances, angles, polygon centroids or areas), and to output a final numeric answer in the same format as the raw setting. While the interpreter improves numeric precision, models can still fail on complex tasks either due to incorrect spatial reasoning or because the generated code is incomplete or erroneous. Results are reported in Table 9.

**VLM setting and renderings.**  We also evaluate VLM inputs by providing a top-down rendered floorplan image alongside the text question. We use two controlled rendering styles derived from the same symbolic layout: (i) a minimal contour-based rendering, where room and object polygons are shown as boxes with text labels (similar to Fig. 3, right), and (ii) an icon-based rendering, where objects are depicted with furniture icons instead of bare contours (similar to Fig. 1). Because HSSD contains a wider variety of object categories than our current icon library, icon-based renderings are used only for synthetic layouts, while HSSD layouts use the contour style. VLM results under these input conditions are summarized in Table 9.

Table 9 shows that Code Interpreter substantially improves metric-dominated scalar tasks (distance, angle, visibility, placement), but offers limited benefit on planning-heavy tasks (Max Box, Shortest Path), indicating that remaining errors are primarily spatial rather than arithmetic. VLM inputs provide selective gains, mainly on generated layouts and for object-fit or metric cues, while not consistently improving global performance.

Table 9: Task accuracy for `gpt-4.1-2025-04-14` and `gpt-4.1-mini-2025-04-14` under three settings: raw text-only input (Raw), tool-augmented reasoning with a Python interpreter (Tools), and vision-language input using rendered floorplans (Img; Icons). **HSSD** columns report performance on 200 human-designed scenes. **Generated** columns report performance on 200 synthetic scenes (50 kitchens, 75 living rooms, and 75 bedrooms). Cell colors indicate gains (green) or drops (red) relative to the corresponding Raw setting for the same model and task.

| | HSSD | | | | | | Generated | | | | | | | |
| | gpt-4.1 | | | gpt-4.1-mini | | | gpt-4.1 | | | | gpt-4.1-mini | | | |
| Question | Raw | Tools | Img | Raw | Tools | Img | Raw | Tools | Img | Icons | Raw | Tools | Img | Icons |
|---|---|---|---|---|---|---|---|---|---|---|---|---|---|---|
| Pair distance | 56 | 99 | 60.5 | 37 | 98 | 51.5 | 95.5 | 99.5 | 99 | 99.5 | 86.5 | 98.5 | 94.5 | 99.5 |
| Placement | 64.5 | 95 | 70.5 | 76.5 | 87.5 | 78.5 | 65 | 92.5 | 61.5 | 71.5 | 76 | 95.5 | 70.5 | 81 |
| Repositioning | 47 | 83.5 | 44.5 | 40.5 | 68 | 37 | 56.5 | 48 | 42 | 39 | 70 | 59.5 | 45 | 50.5 |
| Free space | 16 | 44 | 21.5 | 15.5 | 42 | 26 | 41 | 28.5 | 55 | 45 | 24.5 | 27.5 | 26 | 22.5 |
| Visibility | 46.5 | 86.5 | 36.5 | 52 | 70 | 37 | 80 | 89 | 61.5 | 56.5 | 88 | 71.5 | 72.5 | 68 |
| View angle | 55 | 96 | 63.5 | 46 | 93 | 54 | 92 | 99 | 93.5 | 95.5 | 91.5 | 97 | 92.5 | 92.5 |
| Max box | 7.5 | 3 | 4 | 4.5 | 3.5 | 4 | 12 | 11.5 | 12 | 10.5 | 11 | 8 | 7.5 | 9.5 |
| Shortest path (valid) | 25 | 12.5 | 22.5 | 15 | 14 | 16.5 | 51 | 34 | 46 | 52 | 46.5 | 24.5 | 44 | 45 |
| Shortest path (Fréchet) | 22.5 | 12.5 | 23.5 | 12.5 | 9.5 | 13.5 | 40.5 | 31 | 44 | 45 | 24.5 | 21 | 33 | 27 |

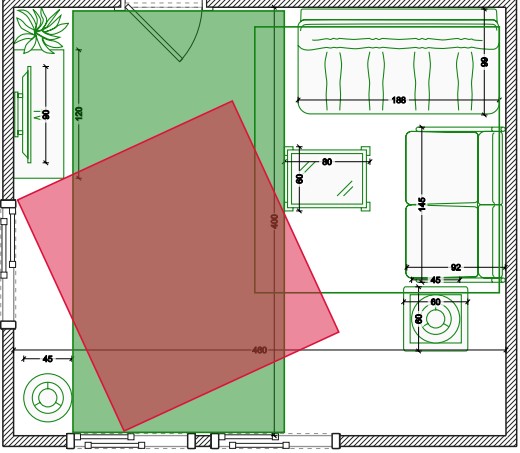
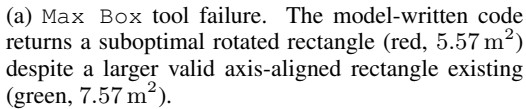
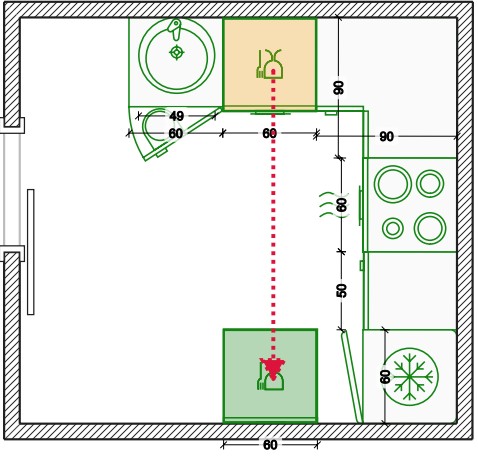

(a) `Max Box` tool failure. The model-written code returns a suboptimal rotated rectangle (red, $5.57\,\mathrm{m}^2$) despite a larger valid axis-aligned rectangle existing (green, $7.57\,\mathrm{m}^2$).

(b) `Repositioning` tool failure. The true maximum downward move is $1.96\,\mathrm{m}$, but the model's code outputs 0 by treating boundary contact as collision.

Figure 7: Representative failure cases in the tool-augmented setting. While the python interpreter improves numeric precision, complex tasks can still fail due to imperfect model-written code and geometric edge cases.

As illustrative failure cases, we also observe that tool augmentation does not guarantee correctness for tasks that require precise geometric reasoning. In `Max Box` (Fig. 7a), the model-generated Python program performs an approximate rotated/grid search and returns a suboptimal rectangle (red, $5.57\,\mathrm{m}^2$), despite a larger valid axis-aligned solution existing (green, $7.57\,\mathrm{m}^2$). In a second example from `Repositioning` ( Fig. 7b), the ground-truth maximum downward displacement of the dishwasher is $1.96\,\mathrm{m}$, but the model's tool-based code predicts zero movement. Inspection shows that the program treats a shared boundary (the dishwasher touching another object or wall) as a collision, and therefore incorrectly concludes that no valid motion is possible for this concrete task. These cases highlight that tools reduce arithmetic imprecision, but complex tasks may still fail due to imperfect model-written algorithms and sensitivity to geometric edge cases (e.g., boundary contact vs. overlap).

# G SENSITIVITY STUDIES FOR NUMERIC TOLERANCES, PATH THRESHOLDS AND ADDITIONAL ABLATIONS

## G.1 SENSITIVITY STUDIES FOR NUMERIC TOLERANCES.

We examine the robustness of our numeric evaluation tolerances for both scalar and area-based question types. For clarity and cost control, we conduct these sensitivity analyses on a representative subset of six models (three reasoning-focused and three general-purpose).

For scalar metrics (e.g., Pair distance, View angle, Repositioning, and Max Box), Figure 8 shows that relative errors are sharply concentrated near zero across models, with a clear knee before $\varepsilon_{\mathrm{rel}} = 2\%$ (red dashed line). The accompanying tolerance sweep in Figure 10 (left) demonstrates smooth, monotone accuracy gains while changing tolerance from $0.5\%$ to $5\%$ without changing model rankings, indicating that the selection of $2\%$ lies in a stable regime rather than being outcome-sensitive.

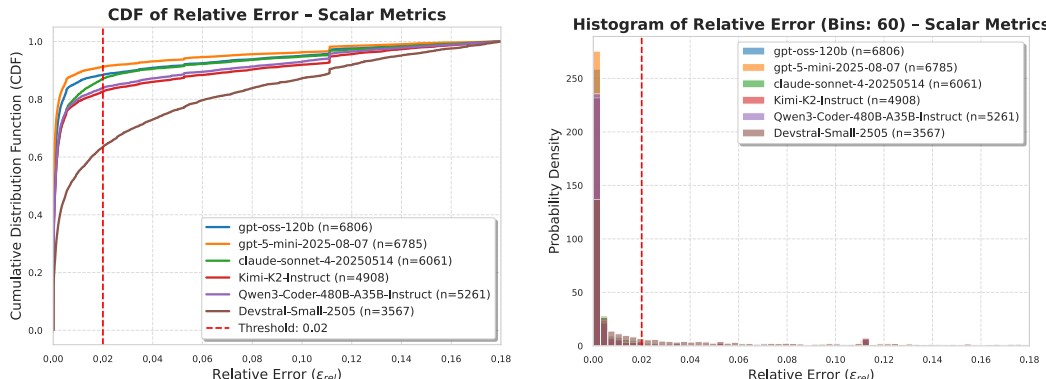

Figure 8: Aggregated relative-error distributions for scalar metrics across models. Left: CDF of relative error; the dashed line marks $\varepsilon_{\mathrm{rel}} = 2\%$. Right: Histogram showing a strong peak near zero and a thin long tail.

For free-space (area) questions, Figure 9 shows a broader, heavier-tailed error distribution; nevertheless, $5\%$ tolerance lies near saturation for higher-performing models while remaining strict for weaker ones. The sweep in Figure 10 (right) confirms that relaxing area tolerance from $1\%$ to $10\%$ yields gradual improvements and preserves qualitative conclusions.

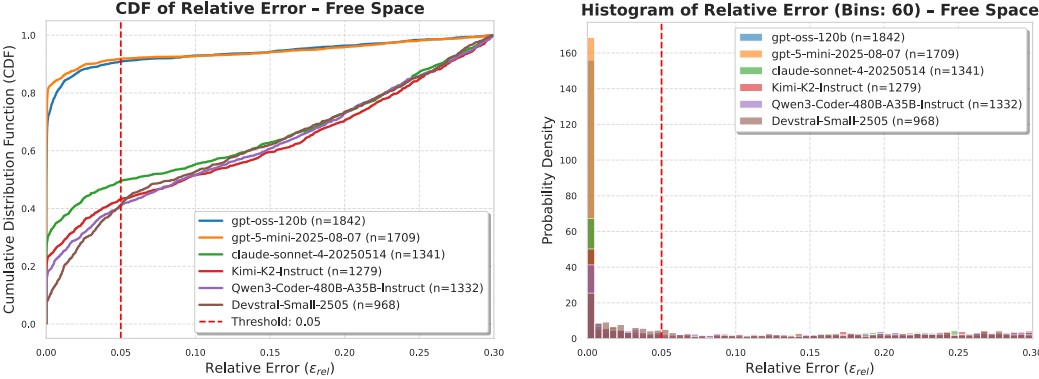

Figure 9: Aggregated relative-error distributions for free-space (area) questions across models. Left: CDF of relative error; the dashed line marks $\varepsilon_{\mathrm{rel}} = 5\%$. Right: Histogram showing broader, heavier-tailed errors compared to scalar tasks.

Overall, these results justify our use of a $2\%$ tolerance for scalar outputs and a $5\%$ tolerance for complex area computations.

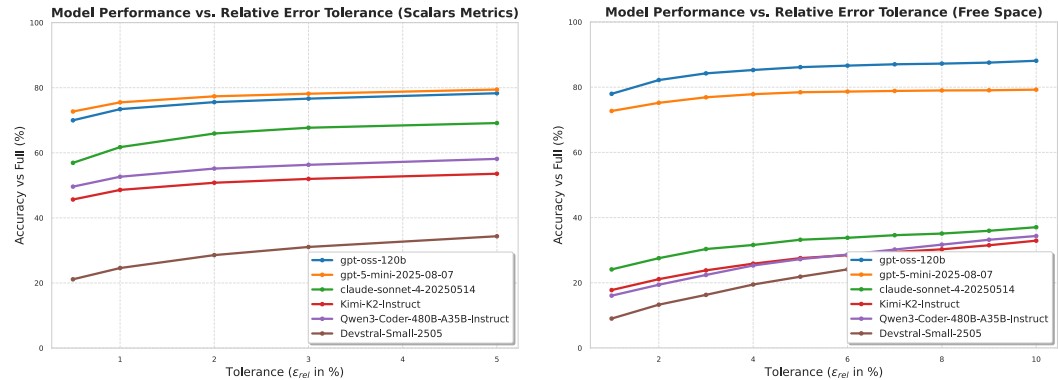

Figure 10: Tolerance sweeps for scalar (left) and free-space area (right) tasks. Aggregated accuracy increases smoothly as tolerance is relaxed (0.5–5% for scalars; 1–10% for areas), while model rankings remain unchanged.

### G.2 SENSITIVITY STUDIES FOR PATH THRESHOLDS

We analyze robustness of shortest-path scoring with respect to the Fréchet threshold $\tau$. Figure 11 shows that accuracy increases smoothly as $\tau$ is relaxed, and model rankings remain almost stable across the sweep. We therefore choose $\tau = 0.6\,\text{m}$ as deviations within roughly $0.6\,\text{m}$ correspond to paths that remain traversable for a person and allow minor alternate routes without accepting qualitatively different solutions.

In addition, path validity requires collision-free traversal under a clearance buffer of $0.15\,\text{m}$. This value represents a minimal safety margin; larger buffers (e.g., $0.3\,\text{m}$) would incorrectly invalidate many feasible paths in compact rooms, particularly around $\sim 0.6\,\text{m}$-scale kitchen utilities, and would over-penalize narrow but realistic layouts.

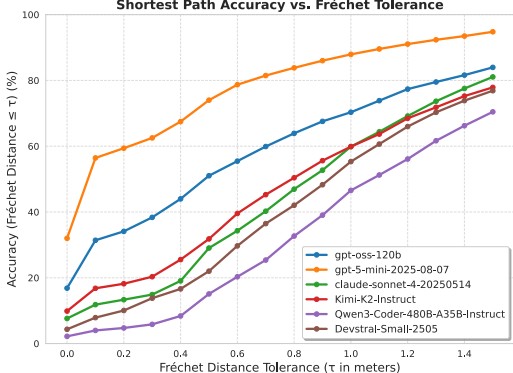

Figure 11: Shortest-path accuracy as a function of Fréchet tolerance $\tau$. Accuracy rises smoothly with increasing tolerance, while model ordering changes slightly. We use $\tau = 0.6\,\text{m}$ as a moderate, human-scale tolerance.

### G.3 PROMPT SENSITIVITY ABLATION

To assess model robustness to prompt variation, we conducted an ablation in which each question was regenerated using the same template but with alternate object references. For example, a question originally referring to a "sofa" might instead use a "bookshelf" in the regenerated version. This allowed us to evaluate whether model performance remains stable under changes in object content while preserving linguistic structure.

As shown in Table 11, accuracy under prompt regeneration is broadly stable; larger models change little, and smaller ones fluctuate modestly. We observe somewhat higher sensitivity for `Repositioning`: paraphrases can implicitly select different target objects or motion directions, occasionally introducing additional complexity or non-movable cases. Overall, the evaluation appears robust to prompt-level variations.

Table 10: Accuracy under prompt variation. Each cell displays performance on the original prompt, followed by a regenerated version with alternative object references.

| Model | Repositioning | View Angle | Visibility |
|---|---|---|---|
| GPT-OSS-120B | $60.5 \rightarrow 59.0$ | $74.0 \rightarrow 80.0$ | $70.0 \rightarrow 77.0$ |
| GPT-OSS-20B | $40.0 \rightarrow 50.0$ | $37.5 \rightarrow 35.5$ | $45.5 \rightarrow 50.0$ |
| Qwen3-235B-A22B | $39.0 \rightarrow 49.0$ | $50.5 \rightarrow 48.0$ | $70.5 \rightarrow 74.0$ |

### G.4 SEMANTIC ABLATION

To evaluate the model's reliance on semantic object labels rather than purely geometric reasoning, we conduct a *semantic ablation* experiment. In this setting, object identifiers in the scene are permuted (e.g., swapping `bed` and `chair` labels) while keeping all geometry unchanged. The regenerated prompts thus refer to the same physical configuration but with altered object semantics, for instance, a question originally phrased as "move the `chair` left" becomes "move the `bed` left," even though the underlying geometry is identical.

As summarized in Table 11, tasks that rely primarily on pure metric computation or topological cues, such as `View Angle` and `Visibility`, show minimal changes in accuracy, with only small fluctuations due to prompt phrasing. However, performance on the action-based `Repositioning` task drops sharply under semantic perturbation, indicating that the model partially grounds its reasoning in object semantics rather than spatial configuration alone. This suggests that large language models may entangle linguistic priors with geometric inference when tasks involve physical movement or interaction.

Table 11: Accuracy under prompt variation. Each cell displays performance on the original prompt, followed by a regenerated version with alternative object references.

| Model | Repositioning | View Angle | Visibility |
|---|---|---|---|
| GPT-OSS-120B | $60.5 \rightarrow 40.0$ | $74.0 \rightarrow 73.0$ | $70.0 \rightarrow 77.0$ |
| GPT-OSS-20B | $40.0 \rightarrow 28.0$ | $37.5 \rightarrow 35.5$ | $45.5 \rightarrow 44.5$ |
| Qwen3-235B-A22B | $39.0 \rightarrow 37.0$ | $50.5 \rightarrow 43.0$ | $70.5 \rightarrow 74.0$ |

## H CASE STUDIES BY QUESTION TYPE

To better understand the sources of model failure, we conducted a qualitative analysis of representative examples from the benchmark. This section presents visualizations of selected test layouts alongside model responses. By examining both correct and incorrect outputs, we aim to identify common failure patterns and reasoning bottlenecks across different architectures.

### H.1 PAIR DISTANCE

**Task Definition**
In this task, the model is asked: *"Calculate the euclidean distance between the centroids of the `obj_1` and the `obj_2`."* In the visualization in Figure 12, these two polygons correspond to the sink and the shower; the goal is to compute their centroid-to-centroid distance.

**Ground-Truth Computation**
To establish the correct answer, we first compute the centroid $(x, y)$ of each polygon. This is done using the *shoelace formula*, which calculates the centroid based on the polygon's vertices. Once

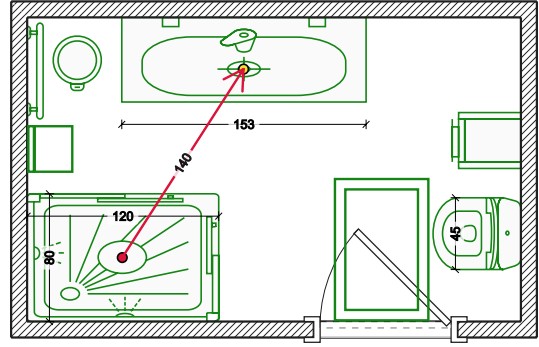 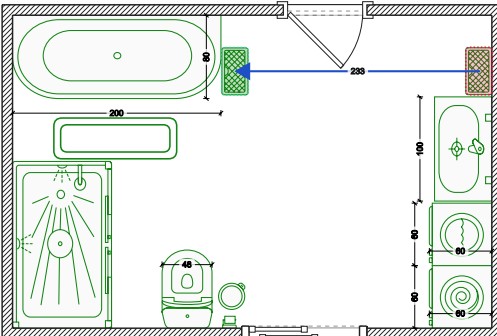

Figure 12: Pair Distance (bathroom): the red line indicates the centroid-to-centroid segment.

Figure 13: Repositioning (bedroom): red fill indicates the initial pose of `bin_2` and green fill indicates the final pose.

both centroids are obtained, the *Euclidean distance* between them is computed. A predicted answer is considered correct if it falls within a tolerance of $2\%$ of the ground-truth distance.

**Main Issue**

Models often fail on the HSSD dataset because they compute the centroid incorrectly. In earlier experiments, some models used the center of mass instead of the centroid; therefore, the word *centroid* is now explicitly stated in the prompt. Almost all wrong answers come from calculation mistakes in the centroid formula (areas, sums, divisions), not from the distance step. This error does not depend on polygon complexity (number of vertices).

H.2 REPOSITIONING

**Task Definition**

We pose the question: *"Calculate how far the `object` can be moved in the `direction` before it touches another object or the wall."* In the visualization in Figure 13, the object of interest is `bin_2` and the direction is leftward; the goal is to compute its maximum leftward translation before contacting a bathtub.

**Ground-Truth Computation**

Simulate a leftward, axis-aligned slide of `bin_2`. Advance until the next step would overlap a wall or another object; take the last non-overlapping pose. Measure the travel distance from the initial position to that pose. Accept model answers within $2\%$ tolerance.

**Main Issue**

Narrow gaps and obstacles with arbitrary orientations make clearance difficult to estimate. Models often fall back to axis-aligned bounding boxes (discarding shape orientation) or omit the obstacle-union step, which leads to systematic over- or underestimation of feasible travel distances in any direction (left, right, up, or down).

H.3 FREE SPACE

**Task Definition**

We consider the following question: *"Calculate the total non-occupied floor area in the `room`?"* In the visualization in Figure 14, the mint-highlighted area represents the space that remains free of objects within the game room; the goal is to compute its total area.

**Ground-Truth Computation**

We compute the free floor area using geometric operations provided by `Shapely` (Gillies et al., 2007). All object polygons within the room are merged using `unary_union` to correctly handle overlaps. The occupied area is then obtained from this union, and the free area is computed as

$$A_{\text{free}} = A_{\text{room}} - A_{\text{union(objects)}}.$$

A model prediction is considered correct if it falls within $5\%$ of the ground-truth free area.

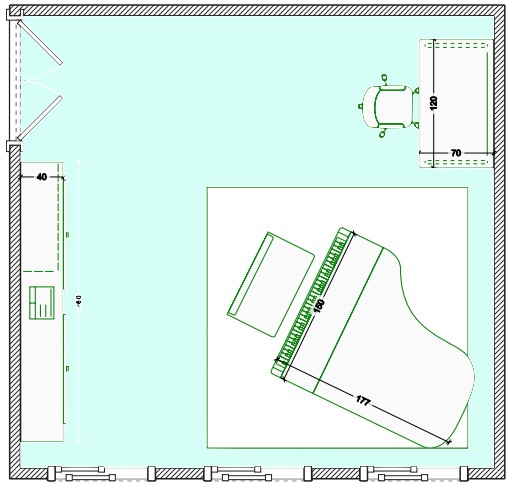
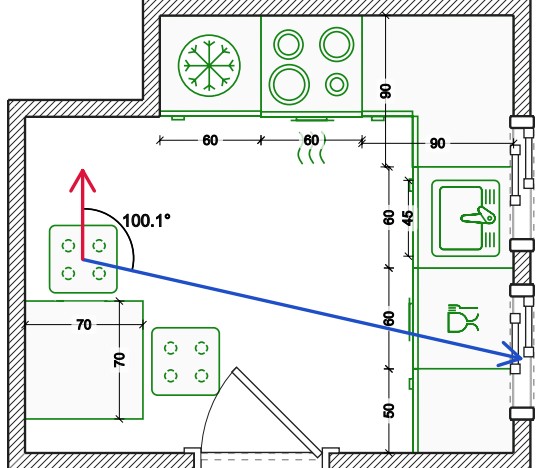

Figure 14: Unoccupied Floor Area (game room): the mint fill indicates the unoccupied (free-space) region within the game room.

Figure 15: View Angle (kitchen): smallest absolute angle between the vector from the centroid of the chair_2 to the centroid of the window_1 and global north $(0, 1)$.

**Main Issue**

Accurate free-space estimation hinges on correctly handling overlapping obstacles. A common failure mode is to subtract each object's area independently, rather than forming their geometric union, which double-counts overlaps and systematically underestimates available area. For instance, when we evaluate on *HSSD* layouts using *GPT-OSS-120B*, cumulative accuracy declines as object count and overlap increase (see Figure 16), consistent with this union-omission error.

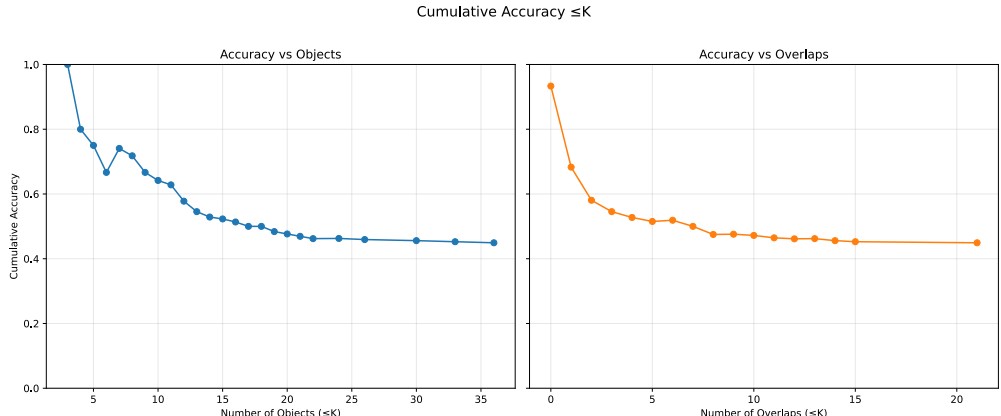

Figure 16: Cumulative accuracy versus layout complexity for HSSD using GPT-OSS-120B on the Free-Space task. Accuracy declines sharply as object count and overlap increase, reflecting the model's difficulty in handling overlapping geometries.

## H.4 VIEW ANGLE

**Task Definition**

The prompt is: *"Compute the smallest absolute angle between the vector from the centroid of* obj_1 *to the centroid of* obj_2 *and the global north vector* $(0, 1)$*; report $\theta$ in degrees."* In the visualization in Figure 15 (kitchen), obj_1 is the chair_2 and obj_2 is the window_1; the goal is to return $\theta$, judged correct within a 2% tolerance.

**Ground-Truth Computation**

(1) Compute centroids $\mathbf{c}_s$ (for `sofa`) and $\mathbf{c}_v$ (for `TV`) using the polygon shoelace formula.

(2) Form the displacement vector $\mathbf{d} = \mathbf{c}_v - \mathbf{c}_s$ and its unit vector $\hat{\mathbf{d}} = \dfrac{\mathbf{d}}{\|\mathbf{d}\|}$.

(3) Let the global north vector be $\mathbf{n} = (0, 1)$ (already unit length). Compute the cosine via $\cos\theta = \text{clip}(\hat{\mathbf{d}} \cdot \mathbf{n}, -1, 1)$.

(4) Convert to degrees and take the smallest absolute angle: $\theta = \arccos(\cos\theta) \cdot \dfrac{180}{\pi} \in [0°, 180°]$.

A prediction is correct if it is within $2\%$ of the ground-truth angle $\theta$.

**Main Issue**

On HSSD layouts, most errors come from centroid *calculation* mistakes (areas/sums/divisions in the shoelace step), not from the dot product. To avoid ambiguity, the prompt explicitly says *centroid*. On synthetic layouts (4-point, axis-aligned boxes), centroids are trivial, and this issue does not appear.

## H.5 PLACEMENT

**Task Definition**

We consider the following question: *"Check if a `given object` can be placed in the room without overlapping walls or other objects?"* In the visualization in Figure 17, the object is a $2.5\,\text{m} \times 1.0\,\text{m}$ antique storage chest and the room is the living room; the goal is to determine whether a collision-free placement is possible. This task evaluates collision detection, spatial constraints, and free-space reasoning.

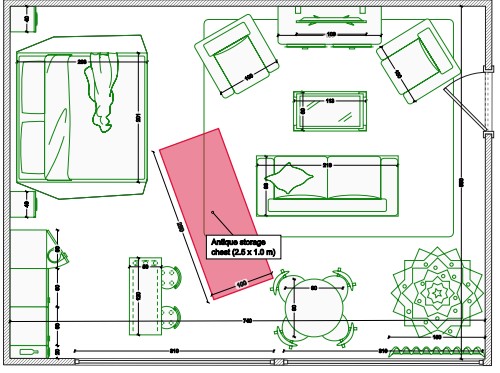

Figure 17: Placement (living room): determine whether an *antique storage chest* can be placed without overlapping walls or existing objects (collision-free feasibility).

Figure 18: Shortest Path (living room): ground-truth shortest walkable path with $15\,\text{cm}$ clearance is shown in purple; the model's predicted path in red intersects `armchair_1`, illustrating a failure case for `GPT-OSS-120B` due to incorrect obstacle/clearance handling.

**Ground-Truth Computation**

Form the room polygon and the union of all existing object polygons. Allow arbitrary rotation (non–axis-aligned) for the $2 \times 1.5\,\text{m}$ rectangle. Search over poses: for each orientation $\theta$, test placements where the rotated rectangle is strictly inside the room polygon and has no intersection with the object union (i.e., *contains* check for the room and *disjoint* check for obstacles). If any collision-free pose exists, return `True`; otherwise `False`. Compare the model's Boolean prediction to this result.

**Main Issue**

The task is harder when non–axis-aligned placements are allowed. Models often mis-handle overlap checks under rotation and falsely report feasibility/infeasibility due to incorrect intersection computations.

## H.6 SHORTEST PATH

**Task Definition**

The question asks: *"Determine the shortest valid path that maintains a clearance of $d$ cm from all other objects, starting from centroid of the* `obj_1` *and ending at the centroid of the* `obj_2`*."* In Figure 18 (living room), `obj_1` is the `TV`, `obj_2` is `armchair_2`, and $d = 15$ cm; the goal is to compute the minimum-length collision-free path (and its length). The figure shows a failure case for `GPT-OSS-120B`, where the predicted path violates the clearance by intersecting the `armchair_1`.

**Ground-Truth Computation**

Offset obstacles (equivalently, erode free space) by $0.15$ m to enforce clearance. Run A\* on the navigable grid to obtain the shortest collision-free path polyline between `TV` and `armchair_2`. A model path is valid if it is collision-free under the same clearance; it is judged correct if its Fréchet distance to the ground-truth path is $\leq 0.6$ m.

**Main Issue**

More objects and overlaps make clearance buffering, merge obstacles, and narrow corridors, increasing failure modes. Models often mishandle overlaps, producing paths that cut through obstacles or declaring no path when one exists.

## H.7 VISIBILITY

**Task Definition**

The prompt is: *"Find all objects that intersect the vector from the centroid of the* `obj_1` *to the centroid of the* `obj_2`*."* In the visualization in Figure 19 (office), `obj_1` is the `window` and `obj_2` is the `bin`; the goal is to return the set of objects that intersect this segment (excluding the endpoints).

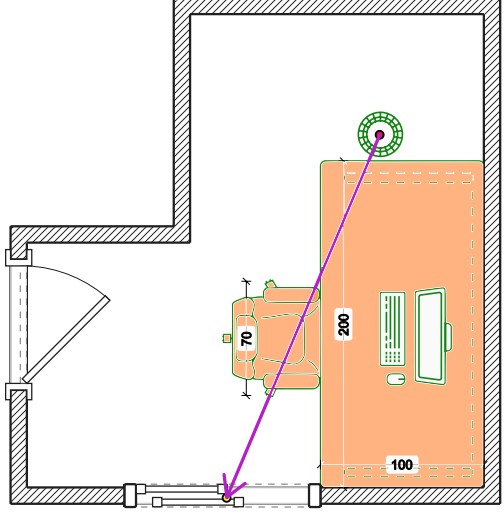 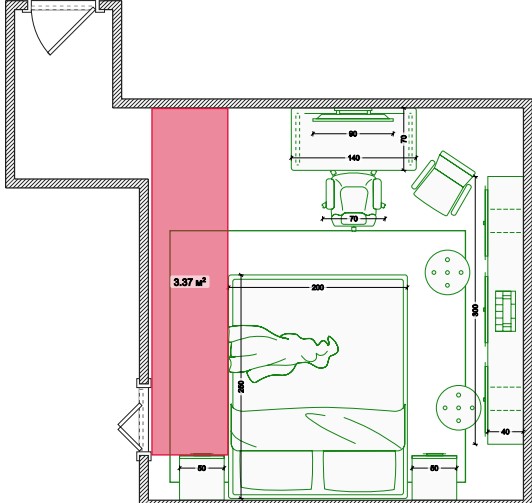

Figure 19: Visibility (office): the `table` and `armchair` (highlighted in orange) intersect the line segment from the centroid of the `window` to the centroid of the `bin` and constitute the correct answer.

Figure 20: Max Box (bedroom): the red rectangle shows the largest rectangular region that can be placed without overlaps, excluding soft coverings such as rugs.

**Ground-Truth Computation**

Compute the centroids of `window` and `bin`; form the line segment between these centroids. Return the set of objects whose bounding boxes intersect this segment, excluding endpoint touches (i.e., ignore cases where the segment only touches at its endpoints).

**Main Issue**

On HSSD layouts, accuracy is slightly worse due to the larger number of objects and overlaps; the increased number of polygons along the line increases intersection ambiguity and error rates.

## H.8 MAX BOX

**Task Definition**

We consider the following question: *"Calculate the area in square meters of the largest rectangle that can fit inside the room"*. In the visualization in Figure 20 (bedroom), the goal is to compute the maximum-area non–axis-aligned rectangle that fits without overlapping any obstacles, and to report its area.

**Ground-Truth Computation**

Let $R$ be the room polygon and $O$ the union of all object polygons except rugs (soft coverings). Compute free space $F = R \setminus O$. Search over orientations $\theta \in [0, \pi)$: rotate $F$ by $-\theta$, find the largest *axis-aligned* empty rectangle inside the rotated $F$, record $(w_\theta, h_\theta)$ and area $A_\theta = w_\theta h_\theta$, then map back to get $(w^*, h^*, \theta^*)$ with $A^* = \max_\theta A_\theta$.

**Main Issue**

Harder than simple placement: the model must *optimize* size and orientation, not just answer yes/no. Allowing rotation makes the search non-convex; more objects and overlaps increase combinatorial complexity. Models often (i) ignore rotation and return an axis-aligned box, or (ii) mis-handle overlaps in free space, leading to under- or over-estimated maxima.

## I PROMPTS

This section illustrates the design of prompt templates used in our benchmark. We first show a representative example of a question prompt, demonstrating how natural language templates are instantiated to elicit spatial reasoning skills (Figure 21).

Next, we present two examples of layout-generation prompts for bedrooms. The first specifies the creation of base room boundaries and openings (walls and windows) (Figure 22). The second demonstrates how furniture and objects are placed within the generated layout to yield a complete scene (Figure 23).

For completeness, full prompt templates, formatting rules, and implementation details are provided in the supplementary code to support reproducibility.

**Prompt: Free Space**

Given the {room_type} layout in {format}, calculate the total non-occupied (free) floor area in square meters ($m^2$).

Room layout: {room}

Begin with printing a concise checklist (3–7 bullets) of the conceptual steps necessary for calculating the free space. Then, carefully walk through each reasoning step required to calculate the area.

If the format, object names, or required input data are missing, invalid, or inconsistent, reply with: `*Final answer*:  ERROR`

Limit your output to the step-by-step reasoning only, and do not include any internal reasoning unless explicitly requested. Clearly state the final answer on the last line using the exact format specified below.

```
### Output Format
<step-by-step calculations>
*Final answer*:  <area>
```

Where `<area>` is a float rounded to three decimal places, representing the free area in $m^2$. For example: `*Final answer*:  12.347`

Figure 21: Prompt for computing the largest empty rectangle area within a room layout using Chain-of-Thought reasoning.

**Prompt: Generate Bedroom Layouts**

Generate a dataset of {N} bedroom layouts in JSON format. Each layout must include:

- A unique `layout_id`
- A `room` dictionary with:
    - `width`, `depth`, `units` (meters)
    - `shape` ("rectangular", "L-shaped", or "open")
    - `shape_description`, `intended_use`, and `bed_size_suggestion`
- An `objects` list with dictionaries containing:
    - `label`, `bbox` [y0, x0, y1, x1], and a descriptive `comment`

Layouts must obey structural and spatial constraints:

- 50% rectangular, 30% L-shaped, 20% open.
- L-shape cutouts in corners; each remaining segment $\geq 1.5$ m.
- All layouts must include a door (except open types); avoid placing doors and windows on the same short wall.
- Windows must span >15% of usable floor area, with equal sizing on shared walls and valid grouping logic.
- Optional elements: fireplace (for master bedrooms), closet alcove.
- No overlap or out-of-bound placement. Fireplace must not overlap with doors/windows.
- Follow a consistent coordinate system: top-left origin, x=width (left to right), y=depth (top to bottom).

Return a JSON list of {N} valid layouts. No comments or trailing metadata.

Figure 22: Summarized data generation prompt for producing structured and constrained bedroom layouts.

**Prompt: Fill Bedroom Layout with Objects**

Given a predefined bedroom layout style and room geometry in JSON format, generate a filled 2D bird-view layout. Include a list of placed objects with their bounding boxes and explanatory comments.

Essential fields:

- Each object must have a `"label"`, `"bbox"` ([y0, x0, y1, x1]), and a descriptive `"comment"`.
- Furniture labels include: `"bed"`, `"nightstand"`, `"dresser"`, `"wardrobe"`, `"desk"`, `"chair"`, `"armchair"`, `"rug"`, `"lamp"`, etc.
- Architectural elements (`"door"`, `"window"`, `"cutout_area"`, `"fireplace"`, `"closet_alcove"`) must match the input layout and remain unmodified.

Placement priorities:

1. Place the `"bed"` according to the `bed_size_suggestion` and layout style.
2. Add essential storage: `"dresser"`, `"wardrobe"`, or use `"closet_alcove"` if defined.
3. Add secondary items (e.g., `"nightstand"`, `"desk"`, `"chair"`) only if space and clearance allow.
4. Add decorative or optional items (`"rug"`, `"mirror"`, `"floor_lamp"`, `"plant"`) last.

Constraints:

- Maintain at least 0.75 m clearance for walkways and door swing.
- Beds require 0.6–0.75 m of access space on sides and foot (unless against wall).
- Wardrobes/dressers need 0.6–0.8 m clearance for drawer/door use.
- No object overlap (except table lamps on nightstands or rugs under furniture).
- Use walls efficiently; avoid blocking windows unless unavoidable.
- Ensure mirror has 0.75 m clearance in front; treat `"rug"` as an anchor but optional.

Final output: a JSON list of objects, including placement and comments. No layout geometry should be altered.

Figure 23: Prompt for populating a bedroom layout with functionally and spatially valid object placements, following layout-specific design rules.

# J SUPPLEMENTARY ACCURACY ANALYSES

Table 12: % token-limit stop reason for **general models**.

| Question | Room | claude sonnet-4 | gpt-4.1 | Kimi-K2 Instruct | Qwen3 Coder-480B | Qwen3 235B | gpt-4.1 mini | Qwen3 30B | Devstral Small |
|---|---|---|---|---|---|---|---|---|---|
| Pair distance | K | 0.0 | 0.0 | 0.0 | 0.0 | 0.0 | 0.0 | 0.0 | 0.2 |
| | LR | 0.0 | 0.2 | 0.0 | 0.0 | 0.0 | 0.0 | 0.0 | 0.2 |
| | B | 0.0 | 0.0 | 0.2 | 0.0 | 0.0 | 0.2 | 0.0 | 0.8 |
| | HSSD | 0.0 | 0.5 | 0.0 | 0.0 | 17.5 | 8.5 | 3.5 | 1.5 |
| Placement | K | 0.0 | 0.0 | 0.0 | 0.0 | 6.5 | 0.0 | 1.2 | 1.2 |
| | LR | 0.0 | 0.0 | 0.0 | 0.0 | 6.0 | 0.0 | 1.0 | 1.5 |
| | B | 0.0 | 0.0 | 0.0 | 0.0 | 17.2 | 0.0 | 2.2 | 0.5 |
| | HSSD | 0.0 | 0.0 | 0.5 | 0.0 | 6.5 | 0.0 | 1.5 | 4.5 |
| Reposi-tioning | K | 0.0 | 0.2 | 0.0 | 0.0 | 5.7 | 0.0 | 2.3 | 0.2 |
| | LR | 0.0 | 0.2 | 0.0 | 0.0 | 1.7 | 0.0 | 1.0 | 1.0 |
| | B | 0.0 | 0.0 | 0.0 | 0.0 | 7.5 | 0.2 | 0.8 | 1.3 |
| | HSSD | 0.0 | 0.0 | 0.0 | 0.0 | 47.0 | 2.0 | 29.0 | 4.5 |
| Free space | K | 0.0 | 0.0 | 0.2 | 0.0 | 0.2 | 0.0 | 0.3 | 2.5 |
| | LR | 0.0 | 0.3 | 0.0 | 0.0 | 4.0 | 0.0 | 1.0 | 10.2 |
| | B | 0.0 | 0.0 | 0.0 | 0.0 | 4.5 | 0.0 | 0.3 | 6.3 |
| | HSSD | 0.0 | 2.0 | 0.0 | 0.0 | 52.5 | 1.0 | 39.0 | 28.5 |
| Visibility | K | 0.2 | 0.0 | 0.0 | 0.0 | 0.0 | 0.0 | 0.2 | 0.3 |
| | LR | 0.2 | 0.0 | 0.0 | 0.0 | 0.2 | 0.2 | 0.5 | 0.3 |
| | B | 1.0 | 0.0 | 0.0 | 0.0 | 0.2 | 0.3 | 1.3 | 0.7 |
| | HSSD | 0.0 | 0.0 | 0.0 | 0.0 | 11.5 | 2.5 | 11.0 | 0.0 |
| View angle | K | 0.0 | 0.0 | 0.0 | 0.0 | 0.5 | 0.0 | 0.0 | 0.3 |
| | LR | 0.0 | 0.0 | 0.2 | 0.2 | 1.5 | 0.0 | 0.2 | 0.7 |
| | B | 0.0 | 0.0 | 0.0 | 0.3 | 0.5 | 0.0 | 0.5 | 1.2 |
| | HSSD | 0.0 | 0.5 | 0.0 | 0.0 | 26.5 | 3.0 | 17.5 | 2.5 |
| Max box | K | 0.0 | 0.0 | 0.0 | 0.0 | 22.5 | 0.0 | 6.2 | 1.5 |
| | LR | 0.0 | 0.0 | 0.7 | 0.0 | 49.8 | 0.0 | 29.3 | 0.8 |
| | B | 0.0 | 0.0 | 1.5 | 0.0 | 46.5 | 0.3 | 21.8 | 0.8 |
| | HSSD | 0.0 | 0.0 | 0.5 | 0.0 | 45.5 | 0.5 | 20.0 | 3.0 |
| Shortest path | K | 0.0 | 0.2 | 0.5 | 0.0 | 44.3 | 0.0 | 41.5 | 6.7 |
| | LR | 0.0 | 0.0 | 0.0 | 0.0 | 37.8 | 0.2 | 46.3 | 7.8 |
| | B | 0.2 | 0.0 | 0.0 | 0.0 | 40.2 | 0.0 | 49.0 | 9.2 |
| | HSSD | 0.0 | 0.0 | 0.0 | 0.0 | 35.5 | 0.0 | 43.0 | 27.0 |

Table 13: Question-level accuracy on completed answers for **general models**.

| Question | Room | claude sonnet-4 | gpt-4.1 | Kimi-K2 Instruct | Qwen3 Coder-480B | Qwen3 235B | gpt-4.1 mini | Qwen3 30B | Devstral Small |
|---|---|---|---|---|---|---|---|---|---|
| Pair distance | K | 99.8 | 96.5 | 95.7 | 96.8 | 99.2 | 90.7 | 89.5 | 58.9 |
| | LR | 99.5 | 95.2 | 94.2 | 96.8 | 99.7 | 88.5 | 88.8 | 62.4 |
| | B | 99.7 | 96.3 | 93.5 | 96.5 | 99.5 | 88.0 | 85.2 | 60.5 |
| | HSSD | 88.0 | 56.3 | 75.5 | 66.0 | 81.2 | 40.4 | 46.1 | 56.9 |
| Placement | K | 87.8 | 78.0 | 82.2 | 80.3 | 96.4 | 86.5 | 86.5 | 75.0 |
| | LR | 80.5 | 69.0 | 73.8 | 83.2 | 94.9 | 75.2 | 86.5 | 72.9 |
| | B | 68.8 | 59.8 | 67.5 | 70.8 | 92.6 | 68.7 | 78.7 | 56.6 |
| | HSSD | 72.0 | 64.5 | 73.4 | 70.0 | 87.7 | 76.5 | 72.6 | 57.1 |
| Reposi-tioning | K | 73.8 | 63.9 | 48.7 | 45.3 | 88.5 | 66.3 | 75.4 | 14.9 |
| | LR | 79.3 | 60.4 | 56.7 | 64.5 | 92.9 | 76.8 | 72.9 | 25.3 |
| | B | 71.0 | 55.5 | 48.3 | 59.5 | 85.4 | 72.6 | 70.6 | 21.5 |
| | HSSD | 42.0 | 47.0 | 28.0 | 34.0 | 73.6 | 41.3 | 46.5 | 10.5 |
| Free space | K | 97.8 | 93.2 | 83.1 | 84.2 | 95.2 | 95.2 | 84.1 | 67.9 |
| | LR | 0.2 | 14.2 | 2.8 | 1.8 | 3.7 | 1.2 | 0.5 | 3.0 |
| | B | 2.7 | 31.2 | 1.0 | 1.3 | 9.3 | 0.8 | 1.0 | 0.7 |
| | HSSD | 35.0 | 16.3 | 24.0 | 22.0 | 35.8 | 15.7 | 12.3 | 9.1 |
| Visibility | K | 63.3 | 87.7 | 54.5 | 67.0 | 98.3 | 90.2 | 91.6 | 18.6 |
| | LR | 52.8 | 81.7 | 43.3 | 52.2 | 98.5 | 87.0 | 89.1 | 10.0 |
| | B | 58.1 | 74.8 | 41.0 | 54.0 | 96.8 | 86.6 | 87.5 | 10.9 |
| | HSSD | 20.0 | 46.5 | 22.5 | 26.5 | 79.7 | 53.3 | 51.1 | 9.0 |
| View angle | K | 92.0 | 95.3 | 69.8 | 78.8 | 97.5 | 95.8 | 74.7 | 49.8 |
| | LR | 87.7 | 93.2 | 59.8 | 75.3 | 95.4 | 93.0 | 72.3 | 40.8 |
| | B | 88.0 | 90.2 | 60.3 | 72.8 | 95.5 | 91.2 | 77.2 | 36.3 |
| | HSSD | 67.5 | 55.3 | 28.5 | 46.5 | 68.7 | 47.4 | 41.8 | 30.8 |
| Max box | K | 47.2 | 31.8 | 32.2 | 26.9 | 84.5 | 27.5 | 28.2 | 4.6 |
| | LR | 7.8 | 7.0 | 5.1 | 5.7 | 44.5 | 4.5 | 12.5 | 0.5 |
| | B | 5.8 | 6.8 | 7.4 | 5.0 | 54.5 | 4.9 | 9.0 | 1.7 |
| | HSSD | 5.0 | 7.5 | 2.5 | 2.0 | 21.1 | 4.5 | 3.8 | 1.0 |
| Shortest path (valid) | K | 59.2 | 61.7 | 52.7 | 39.7 | 81.1 | 55.3 | 37.3 | 36.6 |
| | LR | 53.0 | 51.3 | 42.8 | 37.3 | 72.1 | 47.3 | 37.9 | 32.9 |
| | B | 48.6 | 52.2 | 40.7 | 34.7 | 73.8 | 45.8 | 36.3 | 36.9 |
| | HSSD | 28.5 | 25.0 | 18.5 | 18.0 | 40.3 | 15.0 | 9.7 | 14.4 |
| Shortest path (Fréchet) | K | 45.3 | 56.8 | 51.3 | 28.2 | 79.3 | 39.5 | 30.5 | 38.8 |
| | LR | 24.7 | 42.5 | 27.3 | 12.3 | 55.8 | 26.5 | 17.1 | 16.8 |
| | B | 23.0 | 42.2 | 27.7 | 14.2 | 63.5 | 30.5 | 16.0 | 20.2 |
| | HSSD | 15.5 | 22.5 | 18.0 | 8.0 | 31.0 | 12.5 | 4.4 | 15.8 |

Table 14: % token-limit stop reason for **reasoning models**.

| Question | Room | gpt-5 | gpt-oss 120b | DeepSeek R1-0528 | Gemini Flash 2.5 | gpt-5 mini-2025 | gpt-oss 20b | Qwen3 30B Think. |
|---|---|---|---|---|---|---|---|---|
| Pair distance | K | 0.0 | 0.0 | 1.5 | 3.3 | 0.0 | 0.7 | 2.0 |
| | LR | 0.0 | 0.0 | 0.8 | 4.0 | 0.2 | 0.7 | 2.3 |
| | B | 0.0 | 0.0 | 2.5 | 4.3 | 0.2 | 0.7 | 4.7 |
| | HSSD | 0.0 | 11.0 | 70.0 | 86.5 | 67.0 | 39.5 | 72.0 |
| Placement | K | 13.2 | 0.0 | 5.2 | 39.2 | 6.5 | 4.5 | 29.5 |
| | LR | 22.7 | 0.2 | 7.2 | 45.8 | 10.3 | 12.3 | 52.5 |
| | B | 36.2 | 0.2 | 8.2 | 63.3 | 12.8 | 25.7 | 64.5 |
| | HSSD | 27.0 | 0.0 | 6.0 | 84.0 | 16.5 | 15.0 | 70.5 |
| Repositioning | K | 0.3 | 0.0 | 8.3 | 1.0 | 0.0 | 0.8 | 7.2 |
| | LR | 0.0 | 0.0 | 7.5 | 2.2 | 0.0 | 0.5 | 6.7 |
| | B | 0.2 | 0.0 | 3.2 | 1.7 | 0.0 | 0.8 | 8.3 |
| | HSSD | 2.0 | 0.5 | 33.5 | 76.0 | 28.5 | 22.5 | 63.0 |
| Free space | K | 1.2 | 0.0 | 5.5 | 2.0 | 0.0 | 0.8 | 2.3 |
| | LR | 0.2 | 0.2 | 41.7 | 41.7 | 1.7 | 13.7 | 2.3 |
| | B | 0.2 | 0.0 | 12.2 | 30.7 | 1.0 | 7.8 | 3.0 |
| | HSSD | 5.0 | 28.5 | 79.5 | 96.5 | 85.0 | 77.5 | 98.5 |
| Visibility | K | 0.0 | 1.5 | 26.7 | 72.8 | 0.0 | 0.3 | 19.2 |
| | LR | 0.0 | 1.7 | 46.8 | 88.2 | 0.3 | 1.0 | 26.3 |
| | B | 0.0 | 1.2 | 44.5 | 88.3 | 1.5 | 0.2 | 33.3 |
| | HSSD | 0.0 | 5.0 | 87.0 | 99.5 | 56.5 | 29.0 | 96.0 |
| View angle | K | 0.0 | 0.0 | 25.5 | 5.5 | 11.2 | 1.0 | 1.0 |
| | LR | 0.0 | 0.0 | 30.8 | 5.0 | 14.3 | 1.3 | 0.7 |
| | B | 0.0 | 0.0 | 23.8 | 3.7 | 12.7 | 1.2 | 0.8 |
| | HSSD | 0.0 | 7.5 | 84.5 | 75.5 | 73.0 | 38.0 | 66.0 |
| Max box | K | 0.0 | 0.0 | 30.2 | 95.7 | 2.0 | 28.8 | 62.7 |
| | LR | 0.0 | 0.0 | 63.5 | 100.0 | 5.7 | 61.0 | 98.0 |
| | B | 0.0 | 0.0 | 59.3 | 100.0 | 4.7 | 56.0 | 98.2 |
| | HSSD | 0.0 | 0.0 | 58.5 | 100.0 | 22.5 | 33.0 | 99.0 |
| Shortest path | K | 0.2 | 0.5 | 78.5 | 96.2 | 23.5 | 36.2 | 85.2 |
| | LR | 61.7 | 1.2 | 77.8 | 97.8 | 19.8 | 40.3 | 81.3 |
| | B | 0.5 | 0.5 | 79.5 | 98.0 | 17.5 | 50.3 | 83.7 |
| | HSSD | 0.0 | 3.0 | 82.0 | 100.0 | 65.5 | 32.5 | 98.0 |

Table 15: Question-level accuracy on completed answers for **reasoning models**.

| Question | Room | gpt-5 | gpt-oss 120b | DeepSeek R1-0528 | Gemini Flash 2.5 | gpt-5 mini-2025 | gpt-oss 20b | Qwen3 30B Think. |
|---|---|---|---|---|---|---|---|---|
| Pair distance | K | 99.8 | 99.3 | 98.0 | 96.3 | 100.0 | 94.2 | 97.7 |
| | LR | 98.8 | 99.3 | 99.0 | 96.0 | 99.7 | 93.5 | 97.5 |
| | B | 98.3 | 99.5 | 96.8 | 95.5 | 99.7 | 93.8 | 95.3 |
| | HSSD | 69.0 | 78.5 | 25.5 | 12.5 | 32.5 | 40.5 | 18.0 |
| Placement | K | 84.7 | 92.0 | 89.0 | 59.7 | 90.8 | 85.7 | 68.2 |
| | LR | 75.5 | 89.0 | 82.2 | 53.3 | 86.3 | 78.5 | 46.5 |
| | B | 61.2 | 83.5 | 72.0 | 35.8 | 81.2 | 62.0 | 34.5 |
| | HSSD | 70.0 | 85.0 | 79.0 | 16.0 | 75.5 | 74.5 | 28.5 |
| Reposi- tioning | K | 83.0 | 85.5 | 79.2 | 90.5 | 84.5 | 70.8 | 61.5 |
| | LR | 85.5 | 89.8 | 86.2 | 91.2 | 92.8 | 87.8 | 77.0 |
| | B | 77.8 | 83.3 | 83.3 | 85.2 | 84.3 | 78.0 | 69.2 |
| | HSSD | 49.5 | 60.5 | 47.5 | 18.5 | 53.5 | 40.0 | 27.0 |
| Free space | K | 82.5 | 99.0 | 93.0 | 93.3 | 99.5 | 94.8 | 97.0 |
| | LR | 47.0 | 83.3 | 18.3 | 17.7 | 78.5 | 53.2 | 0.0 |
| | B | 50.5 | 87.5 | 34.8 | 33.3 | 82.2 | 74.0 | 1.0 |
| | HSSD | 19.5 | 31.0 | 6.5 | 1.0 | 5.0 | 9.0 | 1.0 |
| Visibility | K | 94.8 | 94.2 | 71.3 | 26.8 | 98.0 | 91.5 | 78.8 |
| | LR | 95.2 | 94.0 | 52.0 | 11.3 | 98.0 | 89.3 | 70.8 |
| | B | 94.2 | 92.5 | 53.5 | 11.2 | 95.5 | 89.2 | 64.2 |
| | HSSD | 57.0 | 70.0 | 10.0 | 0.5 | 39.0 | 45.5 | 3.0 |
| View Angle | K | 96.2 | 98.5 | 73.7 | 92.5 | 88.3 | 93.5 | 98.5 |
| | LR | 93.3 | 97.3 | 68.2 | 91.5 | 84.5 | 92.0 | 98.3 |
| | B | 95.2 | 98.2 | 75.2 | 93.8 | 86.8 | 91.3 | 98.3 |
| | HSSD | 59.5 | 74.0 | 13.5 | 20.0 | 25.5 | 37.5 | 26.0 |
| Max Box | K | 48.5 | 62.8 | 50.5 | 3.7 | 85.2 | 31.3 | 16.7 |
| | LR | 17.3 | 28.2 | 8.0 | 0.0 | 60.3 | 3.8 | 0.8 |
| | B | 13.0 | 30.3 | 11.0 | 0.0 | 61.2 | 5.8 | 0.5 |
| | HSSD | 5.0 | 9.5 | 2.5 | 0.0 | 17.0 | 0.5 | 0.0 |
| Shortest path (valid) | K | 64.7 | 64.2 | 12.3 | 3.2 | 52.3 | 43.0 | 14.2 |
| | LR | 21.2 | 66.0 | 12.5 | 1.5 | 53.7 | 37.7 | 16.7 |
| | B | 58.2 | 57.8 | 7.8 | 1.0 | 52.0 | 30.0 | 14.3 |
| | HSSD | 28.5 | 33.5 | 8.5 | 0.0 | 16.5 | 13.5 | 1.0 |
| Shortest path (Fréchet) | K | 56.3 | 55.5 | 11.5 | 3.2 | 50.5 | 42.2 | 14.0 |
| | LR | 12.3 | 40.5 | 9.3 | 1.3 | 47.5 | 28.5 | 16.3 |
| | B | 39.3 | 44.8 | 6.0 | 0.8 | 47.7 | 24.2 | 14.3 |
| | HSSD | 22.5 | 26.5 | 2.5 | 0.0 | 12.0 | 14.0 | 1.0 |

