# OpenReview forum: "FloorplanQA: A Benchmark for Spatial Reasoning in LLMs using Structured Representations"
_ICLR.cc/2026/Conference — ICLR 2026 Conference Desk Rejected Submission_

### Official Review · Reviewer_FFLW · 2025-10-16

**Soundness:** 2
**Presentation:** 2
**Contribution:** 2
**Rating:** 4
**Confidence:** 3

**Summary:**

FloorplanQA proposes a benchmark for evaluating spatial and geometric reasoning in large language models (LLMs) using structured 2D indoor layouts encoded in JSON/XML, focusing on tasks such as distances, visibility, placement feasibility, free space computation, and path planning with clearance. The dataset comprises 2,000 layouts—1,800 synthetically generated via Gemini 2.5 Pro prompts and 200 layouts extracted from HSSD—paired with 16,000 questions (eight per layout), and the evaluation protocol compares outputs to deterministic ground truth with tolerance thresholds. The study assesses 15 models (7 “reasoning” and 8 “general”) under zero-shot prompts, reports accuracy by model and question, and provides an encoding ablation (JSON vs. XML) that shows limited sensitivity.
Overall, the benchmark highlights that current LLMs can answer simple metric queries but struggle with overlap-heavy layouts, free-space unions, and shortest-path planning, with HSSD layouts exposing more errors than synthetic axis-aligned ones.

**Strengths:**

1. The benchmark isolates symbolic spatial reasoning using structured floorplans rather than images, offering a complementary diagnostic perspective to vision-language tasks
2. A clear taxonomy spans metric, topological, and action-like tasks; answer formats and scoring rules are specified with tolerances for numeric and geometric checks
3. Encoding ablation (JSON vs. XML) suggests limited sensitivity to layout serialization, at least for selected tasks and models

**Weaknesses:**

1. 1800 layouts are generated by an LLM with rule-based filters, and synthetic objects are axis-aligned boxes; only 200 HSSD layouts introduce non–axis-aligned geometry. This raises concerns about realism, diversity, and potential generator biases
2. Questions are posed on single-room layouts; multi-room reasoning and dynamic layout changes (e.g., moving objects and re-evaluating visibility across rooms) are not covered. Room types are mainly kitchens, living rooms, and bedrooms
3. Numeric tolerances (2% for scalars, 5% for complex areas) are asserted but not justified via sensitivity analysis. Path validation thresholds and clearance buffering are not stress-tested across diverse geometries
4. The protocol depends on regex-based parsing with strict “Final answer” markers; while invalid formats are reportedly rare, this design can bias task solvability and error attribution

**Questions:**

How will you mitigate comparability confounds from token budgets and truncation? Can u report per-model truncation rates with error-mode analyses

Could you add sensitivity studies for numeric tolerances (2%/5%) and path validation thresholds (Fréchet, clearance), including stress tests across polygon concavity and overlap density

---

> ### Author Response · Authors · 2025-11-21
>
> We thank Reviewer FFLW for the careful evaluation and constructive feedback. We appreciate your acknowledgement of our task taxonomy, symbolic reasoning focus, and the clarity of our evaluation protocol and scoring rules.
>
> > Could you add sensitivity studies for numeric tolerances (2%/5%) and path validation thresholds (Fréchet, clearance), including stress tests across polygon concavity and overlap density.
>
> We added numeric sensitivity analysis in **Appendix G**. We plotted aggregated CDFs and histograms of relative error for scalar tasks (Figure 8) and for free-space area tasks (Figure 9) using six representative models. These plots show that scalar errors concentrate sharply near zero with a clear knee before 2%, while area errors are broader and heavy-tailed, with 5% lying near saturation for strong models but remaining strict for weaker ones. We also include tolerance sweeps (Figure 10) from 0.5%–5% for scalars and 1%–10% for areas, showing smooth changes in absolute accuracy with stable model rankings. This supports that our chosen 2%/5% tolerances operate in robust, non-sensitive regimes.
>
> We also conducted a sensitivity sweep over the Fréchet-distance threshold for the Shortest Path task (Appendix G.2, Fig. 11). The sweep shows a smooth, monotonic increase in accuracy as threshold is relaxed, while model rankings changes slightly for close models. This indicates that our conclusions are not sensitive to a specific Fréchet threshold. Sweeping clearance would require rerunning all models since it changes the underlying path constraints and ground-truth targets. We fix 0.15 m clearance that adds a realistic margin while remaining non-severe.
>
> > How will you mitigate comparability confounds from token budgets and truncation? Can u report per-model truncation rates with error-mode analyses
>
> We now report per-model truncation rates and a full error-mode breakdown (% truncated, % invalid-format, % valid-but-wrong, % correct) aggregated across question and room types in Table 8, Appendix E. This shows that invalid formatting is rare (<1% on average) and truncation affects mostly big reasoning models, while most failures are valid numeric predictions outside tolerance. As it was posted in discussion with Reviewer cFtC "We treat token budget as a realistic constraint of the task setting, and we also report accuracy on non-truncated answers in Tables 13, 15 (Appendix J) to show quality conditional on completion. However, we do not consider reporting only non-truncated accuracy as fully fair, since truncation occurs disproportionately on the hardest questions. Running fully unbounded-token experiments is an interesting follow-up, but the current fixed-budget setting already reflects realistic deployment conditions and is sufficient to reveal model trends and failure modes."
>
>
> > The protocol depends on regex-based parsing with strict “Final answer” markers; while invalid formats are reportedly rare
>
> Regex-based answer extraction is standard practice in LLM benchmarks such as [GSM8K](https://github.com/openai/grade-school-math?tab=readme-ov-file#solution-extracting) and [MATH](https://huggingface.co/datasets/hendrycks/competition_math), both of which evaluate model outputs by matching a required final-answer token via regular expressions. Our parser accepts mild formatting variations (bold vs non-bold, presence/absence of colons, etc.). In **Table 8** we report the percent of invalid responses aggregated per model; the average is below 1%. We also report % alt accuracy using an alternative parser that ignores “Final answer” markers and extracts the numeric/list value from the last line of the response instead. The difference remains negligible, confirming that our conclusions are not driven by parser design.
>
> > Questions are posed on single-room layouts; multi-room reasoning and dynamic layout changes are not covered. Room types are mainly kitchens, living rooms, and bedrooms.
>
> We agree multi-room and dynamic settings are interesting, but they add a new layer of complexity and would require expanding task definitions and data generation. We believe the current single-room benchmark is already challenging and sufficient to expose systematic patterns in LLM spatial reasoning. We see multi-room extensions as promising future work.
>
> > 1800 layouts are generated by an LLM with rule-based filters, and synthetic objects are axis-aligned boxes; only 200 HSSD layouts introduce non–axis-aligned geometry. This raises concerns about realism, diversity, and potential generator biases
>
> Our results show that LLM-generated layouts follow the same trends as the 200 real HSSD layouts (Figure 2). The synthetic set lets us scale layout diversity in a controlled way, while the real HSSD layouts confirm that the observed model weaknesses persist on realistic geometry. Finally, the symbolic representation is flexible: it can be rendered in multiple visual styles and can support future work on improving realism or fine-tuning.

---

### Official Review · Reviewer_2emM · 2025-10-31

**Soundness:** 3
**Presentation:** 3
**Contribution:** 3
**Rating:** 6
**Confidence:** 3

**Summary:**

The paper proposes a new benchmark, FloorPlanQA, designed to evaluate the ability of large language models (LLMs) to infer spatial awareness and perform reasoning based solely on structured inputs. The benchmark consists of two main components. The first and largest portion is synthetically generated using Gemini 2.5 Pro. The second portion projects 3D scenes from HSSD-200 into corresponding 2D layouts, enriching the dataset with realistic spatial configurations.

To generate questions, the authors employ a template-based approach to form eight question types spanning three categories: Action, Topology, and Metric. This design enables comprehensive evaluationmfrom low-level geometric calculations to higher-level reasoning tasks that require detailed spatial planning.

The experiments involve 15 LLMs, including both naive and reasoning-oriented LLMs, evaluated on the proposed dataset. Results show that current SOTA LLMs still struggle to fully comprehend geometric representations, particularly when layouts involve overlapping spatial elements. The category-wise findings further indicate that low-level tasks are generally easier for models, whereas high-level reasoning tasks remain challenging.

**Strengths:**

- The paper proposes a publicly available dataset for geometric representations of room layouts, enabling the evaluation of LLMs’ spatial and layout understanding.

- An automatic pipeline is introduced to generate synthetic layouts using LLMs, demonstrating strong potential for building scalable benchmarks.

- The proposed benchmark consists of both synthetic and realisitc layouts, providing rich information to be evaluate.

- The proposed benchmark reveals the limitations of current LLMs in comprehending scenes solely from geometric representations and reasoning over them.

- The results show that models struggle with complex layouts but achieve reasonable performance on simpler configurations.

- The paper provides a detailed quantitative analysis of the results, revealing the specific behavior of each LLM across different layout types and task categories.

- The paper also illustrates the consistenl results acorss different types of layout representations (JSON vs XML)

- Paper is well-written and easy to follow. Significant details are provided for reproducibility.

**Weaknesses:**

- The discussion of experimental results appears somewhat shallow (only based on numerical number), with limited analysis of why the models fail on specific tasks. Some failures qualitative analysis would be insightful on model behavior in this task.

- No methods are proposed to improve model performance; even preliminary ideas or directions for enhancement would strengthen the contribution.

- A discussion regarding models with visual training could be valuable, as it may reveal whether such training influences performance on spatial reasoning tasks.

- Although the paper includes examples of actual layouts in the main text, the layouts generated by Gemini are never discussed. Including at least one example would improve understanding and reproducibility.

- The paper mentions a few case studies in the appendix that highlight model mistakes, but these are not referenced in the main content. Briefly mentioning them in the main paper would help readers recognize where to find detailed examples of model failures.

**Questions:**

There are no additional questions, except respond to each weakness raised above.

---

> ### Author Response · Authors · 2025-11-21
>
> We thank Reviewer 2emM for the thoughtful and positive review of our submission. We appreciate your recognition of our dataset design, synthetic–real integration, and the clarity and reproducibility of our experimental analysis.
>
> > The paper mentions a few case studies in the appendix that highlight model mistakes, but these are not referenced in the main content. Briefly mentioning them in the main paper would help readers recognize where to find detailed examples of model failures.
>
> We added an explicit forward reference in the Results section directing readers to the detailed case study visualizations in **Appendix H**.
>
> > The discussion of experimental results appears somewhat shallow (only based on numerical number), with limited analysis of why the models fail on specific tasks. Some failures qualitative analysis would be insightful on model behavior in this task.
>
> We added several concrete failure-case visualizations to the paper. These include examples for the Max Box task (Figure 7a), the repositioning task (Figure 8b), and shortest-path reasoning (Figure 18), along with textual descriptions of other failure modes and references to the corresponding images of ground-truth solutions (Appendix H). Together, these examples illustrate common error patterns such as incorrect geometry interpretation, collision handling, and path planning mistake.
>
> Some cases, such as free-space computation, are harder to visualize directly in an image, while errors in centroid computation for pairwise distance estimation do not require explicit visual illustration, since they are already clear from the symbolic representation and sufficiently explained in the text.
>
> > No methods are proposed to improve model performance; even preliminary ideas or directions for enhancement would strengthen the contribution. A discussion regarding models with visual training could be valuable, as it may reveal whether such training influences performance on spatial reasoning tasks.
>
> We already outlined several future directions in the conclusion. One proposed near-term approach is to hybridize LLMs with external geometric or symbolic tools to compensate for known weaknesses in spatial reasoning. Another direction is to fine-tune models on complex paired symbolic–image floorplan data under explicit spatial constraints.
>
> Moreover, as discussed in response to Reviewer 1riV, we implemented tool-augmented and vision-augmented settings (Appendix F). Tools improve simple metric tasks by reducing arithmetic noise, but do not fully solve harder spatial reasoning failures. Vision provides gains on specific tasks, and these gains may be sensitive to the rendering style.
>
> Beyond these preliminary directions, potential improvements include multi-step interaction (e.g., asking an agent to verify or revise its solution using a rendered floorplan). We believe these represent valuable follow-up projects, and we have added these thoughts as well in the conclusion, while the current paper already provides an important standalone benchmark.
>
> > Although the paper includes examples of actual layouts in the main text, the layouts generated by Gemini are never discussed. Including at least one example would improve understanding and reproducibility.
>
> **Figure 1 (page 4)** explicitly shows three Gemini-generated synthetic layouts (“Generated: kitchen, living room, bedroom”; the first three images) alongside one HSSD layout. The caption directly labels them as generated layouts.
>
> **Sections 3.1 and 3.3** describe the Gemini generation pipeline in detail. Table 1 provides information on room type, internal style, and geometric configuration. A more detailed description is given in Appendix B, and summary statistics (average number of objects, number of overlaps, object density, and area) are reported in Appendix D. Appendix I also contains the prompts used to generate such layouts.

---

### Official Review · Reviewer_cFtC · 2025-11-01

**Soundness:** 3
**Presentation:** 1
**Contribution:** 2
**Rating:** 4
**Confidence:** 4

**Summary:**

FloorplanQA is a benchmark for testing LLMs’ spatial reasoning on indoor floor plans using structured, symbolic layouts (JSON/polygons with rooms, doors/windows, objects, and sizes) rather than images or external tools. It targets three capability groups—metric (e.g., distances, areas), topological (e.g., visibility, occupancy, placeability), and action/path reasoning (e.g., relocation, shortest paths with safety margins). The dataset contains 2,000 layouts (1,800 synthetically generated and 200 derived from HSSD), with 8 questions per layout for a total of 16,000 QA pairs. A unified automatic scoring protocol (numeric tolerances, set matching, geometric validity, Fréchet thresholds for paths) is provided, along with explicit accounting for invalid outputs caused by formatting or truncation, enabling fair comparisons and detailed error diagnosis across models.

**Strengths:**

1. Symbolic input, tool-free setup that isolates pure geometric/topological reasoning without visual noise or help from external solvers.

2. Comprehensive coverage across metric, topological, and action/path tasks, including Free Space, Max Box, Placement, Visibility, and Shortest Path.

3. Strong comparability via automation: strict output formats and tolerance thresholds; tailored scoring rules for numbers, sets, and sequences (e.g., 2–5% tolerances, set matching, Fréchet threshold with collision constraints).

4. Realistic geometric diversity by combining controllable synthetic layouts with irregular polygons from HSSD under a unified representation.

**Weaknesses:**

1. Gap to real-world perception/interaction: purely symbolic floor plans omit imagery, noise, and perception errors, limiting ecological validity for embodied/vision tasks.


2. Planar-geometry focus: limited coverage of richer functional metrics (e.g., door flow, dynamic crowds, reachability and behavior constraints).


3. Sensitive to long-context/token budgets: truncation and formatting issues materially affect outcomes.

**Questions:**

With only symbolic floor plans, this isn’t directly usable. Could you also provide corresponding image-based modeling (e.g., via image-generation models) and even try to generate 3D models?

---

> ### Author Response · Authors · 2025-11-21
>
> We thank Reviewer cFtC for the detailed and constructive assessment of our work. We appreciate your acknowledgement of our symbolic design, broad task coverage, and automated evaluation protocol.
>
>
> > Sensitive to long-context/token budgets: truncation and formatting issues materially affect outcomes.
>
> We added detailed aggregated information in **Table 8, Appendix E**. Our parser fails to extract answers in fewer than 1% of responses in average, indicating that models generally follow the required answer format. We additionally test an alternative parser that extracts the numeric (or list) value on the last line of the output and report the resulting %alt accuracy. This changes accuracy only marginally for a small subset of models.
>
> Truncation mainly affects large reasoning models. We treat token budget as a realistic constraint of the task setting, and we also report accuracy on non-truncated answers in Tables 13, 15, Appendix J, to show quality conditional on completion. However, we do not consider reporting only non-truncated accuracy as fully fair, since truncation occurs disproportionately on the hardest questions.
>
> Running fully unbounded-token experiments is an interesting follow-up, but the current fixed-budget setting already reflects realistic deployment conditions and is sufficient to reveal model trends and failure modes.
>
> > Presentation: 1: poor
>
> We improved presentation substantially. We updated many floorplan figures by replacing bounding boxes with furniture icons, improved figure captions, and added clearer appendix references in the main text for consistency.
>
> > Gap to real-world perception/interaction: purely symbolic floor plans omit imagery, noise, and perception errors, limiting ecological validity for embodied/vision tasks. Planar-geometry focus: limited coverage of richer functional metrics (e.g., door flow, dynamic crowds, reachability and behavior constraints).
>
> We agree these are interesting extensions, but FloorplanQA is designed primarily to support architectural and layout-planning workflows, not robotics perception. Within this scope, symbolic geometry is the correct abstraction and already yields a non-trivial benchmark for spatial reasoning.
>
> Still, we now add VLLM experiments in **Appendix F (Table 9)**, where we accompany the symbolic description with a rendered floorplan image. While images do not help universally, they improve some tasks (e.g., object placement). This reinforces that vision representation matters.

---

> > ### Comment · Reviewer_cFtC · 2025-11-27
> >
> > Thanks for the reply. I acknowledge the effort made during the rebuttal, which partially answered my questions. However, I still do not believe that simple top-down symbolic geometries are enough for scene layout planning.
> > Many existing methods (e.g., LayoutGPT, HOLODECK, LayoutVLM, I-design) represent layouts much better using 3D assets. And such as 'CHOrD: Generation of Collision-Free, House-Scale, and Organized Digital
> > Twins for 3D Indoor Scenes with Controllable Floor Plans and Optimal Layouts' have top-down view and scene render results. Relying solely on top-down plots without considering downstream applications renders the work incomplete in my view.
> >
> > Thus, I will keep my rating.

---

> > > ### Author Response · Authors · 2025-11-28
> > >
> > > We believe the concern reflects a difference in research focus. We note that there are multiple valid directions in this area, such as visual 3D scene generation, understanding images/videos and understanding CAD-like plans. These are independent problems, and our work contributes specifically to the latter. We believe that mastering the processing of simple layouts is a necessary step to solve before moving to highly detailed 3D models. As all above mentioned models still have obvious limitations in floor plan generation part, despite producing beautiful renderings.

---

### Official Review · Reviewer_1riV · 2025-11-02

**Soundness:** 3
**Presentation:** 3
**Contribution:** 3
**Rating:** 4
**Confidence:** 4

**Summary:**

The paper proposes a benchmark to evaluate the spatial understanding and quantitative reasoning capabilities of llms in a indoor floorplan setting. The benchmark describes the geometrical structure and state of a room including the location of doors, windows, structures and obstacles and asks an llm to a set of different questions such as distance between two objects, area of free space, etc. This evaluates how well the llm can internally visualize and reasoning about the spatial context from only the description.

The contribution of the paper is the new benchmark which allows the community to  evaluate llm capabilities in spatial reasoning.

**Strengths:**

- A large dataset containing both synthetic and hand designed samples of structural design schematics in a descriptive json / xml format
- The dataset is well designed  to cover a lot of different cases such as room shapes, object placement, collision, etc.
- Robust evaluation which shows current capabilities of llms for quantitative reasoning in spatial questions.
- Well organized paper describing the data generation and evaluation process

**Weaknesses:**

- The benchmark only evaluates the internal quantitative capabilities and doesn't consider tool use or agentic workflows. LLMs are bad at generating high precision quantitative answers such as "distance between two points". It would be better to see how well the model can plan to reach the objective, but perform the mathematical calculations either using code or external tools like a calculator for better precision.
- The authors haven't evaluated any VLLMs with their benchmark. Since this is a spatial reasoning task, instead of only providing the textual coordinate based description of the room and letting the model decipher the plan, maybe providing a rendered image of the room to a VLLM might improve accuracy. This is an interesting case to see how much performance difference can be gained by providing the rendering of the design.

**Questions:**

- Please provide evidence or results for the items mentioned in weakness.
- In Fig 2 and 3, the right plot containing accuracy by question - which model's accuracy is shown in this plot?

---

> ### Author Response · Authors · 2025-11-21
>
> We thank Reviewer 1riV for the constructive and positive evaluation of our work. We appreciate your recognition of our benchmark design, dataset structure, and the clarity of our methodology and presentation.
>
> > In Fig 2 and 3, the right plot containing accuracy by question - which model's accuracy is shown in this plot?
>
> We updated the caption to clarify that the right images summarizes accuracy by question type, averaged across *all models* within the respective family: general models (top) and reasoning models (bottom).
>
> > Please provide evidence or results for the items mentioned in weakness. 1) The benchmark only evaluates the internal quantitative capabilities and doesn't consider tool use or agentic workflows. 2) The authors haven't evaluated any VLLMs with their benchmark.
>
> We agree these are interesting directions and now provide additional results in Appendix F. Although spatial tasks can be delegated to external tools, the goal of FloorplanQA is to evaluate unaided spatial reasoning from symbolic floorplans, and tool use also introduces a practical cost overhead. Accordingly, we report tool-augmented and vision-augmented (VLM) results, which show only limited gains and indicate that internal spatial and geometric reasoning remains the main challenge.
>
> **Tools**. We turn on the Python interpreter tool and allow the model to write and execute code. Results for GPT-4.1 and GPT-4.1-mini are reported in Table 9, Appendix F. We observe gains on arithmetic-heavy tasks such as pair distance and view angle (e.g. difficulty of which lies in the correct calculation of the centroid), but limited improvement on more complex cases, where the generated code is often correct but implements approximate or incorrect logic, leading to wrong final answers (see Figure 7).
>
>
> **VLLMs**. We also ran VLLMs using rendered floorplan images. Results are reported in the same Table 9, Appendix F. Vision representations can vary across models and training data, and we used two controlled renderings: a simple box-based drawing and a more realistic icon-based rendering, like we use in the paper (in experiments icons are used only for generated layouts since HSSD has wider object variety). We find that images can help for some tasks (e.g., object placement, estimating object centroids), but overall improvements are limited, likely because current VLLMs are not explicitly trained on floorplan-style representations and because the current rendering may not be optimal for their perception.
>
> Finally, our symbolic format is valuable on its own. It can be rendered in multiple ways, enables future fine-tuning with paired text–image data, and opens the door to follow-up work such as floorplan object detection and 3D scene creation.

---

### Author Response · Authors · 2025-12-01
**Discussion summary**

We thank the Area Chair and all reviewers for their thoughtful feedback. We have uploaded a revised paper and supplementary material that include the requested experiments, analyses, and presentation improvements. We appreciate that the reviewers highlighted the strengths of our work, such as dataset design, task taxonomy, and the value of a symbolic spatial reasoning benchmark. Below, we summarize our main updates.

**1. Tool Use and VLM Evaluations (1riV, cFtC).**
We added extensive experiments with Python Interpreter tool-use and VLMs using rendered floorplans (Appendix F).
- **Tools:** Code execution improves arithmetic-based tasks but still fails on geometric reasoning (e.g., Max Box, Shortest Path) due to incorrect spatial logic.
- **VLMs:** Visual renderings help with object placement and distance estimation, but do not reliably solve the benchmark; models still struggle to interpret spatial structure from these complex visual inputs.

These results show that while tools and VLMs are promising extensions, they do not eliminate the core reasoning challenges FloorplanQA is designed to reveal.

**2. Robustness and Evaluation Protocol (FFLW, cFtC).**
We performed sensitivity analyses over numeric tolerances and path validation thresholds (Appendix G) and added error distributions (Fig. 8–11). Our chosen thresholds are stable, and relaxing them changes absolute accuracy but not model ranking. We also report formatting and truncation statistics, and provide alternative parsing results (Appendix E, J), confirming that formatting errors are rare and do not impact the conclusions.

**3. Qualitative Failure Analysis (2emM).**
We added explicit references to Appendix H, which now includes representative failure cases across task categories (e.g., Max Box, Repositioning, Shortest Path), illustrating typical collision and feasibility issues.

**4. Presentation Improvements (cFtC, 2emM).**
We updated all layout visualizations, replacing bounding boxes with furniture icons, revised the captions, and added additional references to the appendix to improve clarity and readability.

**5. Realism and Data Diversity (FFLW, cFtC).**
We clarify that symbolic and CAD-like spatial layouts state a distinct and practically important problem setting, separate from 3D scene perception. Improving reasoning on simple 2D layouts is a necessary step, given that current models still struggle with these tasks, even the simplest ones. We further validate the realism of our synthetic data by demonstrating that performance trends closely match those observed in human-authored HSSD layouts (Fig. 2).

In summary, these revisions address the main concerns raised in the reviews. The updated results state that FloorplanQA is a complete benchmark highlighting persistent spatial reasoning limitations in modern LLMs, even when aided by tools or visual inputs. We hope this work supports continued progress towards geometry-aware language models and floorplan understanding.

---

### Note · Program_Chairs · 2026-01-17
**Submission Desk Rejected by Program Chairs**

The following references in this submission do not refer to real documents and/or have major errors in bibliographic information:

 "Weidong Di, Kejia Liu, Xuefei Ma, and Yongdong Dong. Deep layout of custom-size furniture for interior decoration. IEEE Transactions on Visualization and Computer Graphics, 27(9):38593870, 2020.